# Synergies Between Disentanglement and Sparsity: a Multi-Task Learning Perspective

## Abstract

Although disentangled representations are often said to be beneficial for downstream tasks, current empirical and theoretical understanding is limited. In this work, we provide evidence that disentangled representations coupled with sparse base-predictors improve generalization. In the context of multi-task learning, we prove a new identifiability result that provides conditions under which maximally sparse base-predictors yield disentangled representations. Motivated by this theoretical result, we propose a practical approach to learn disentangled representations based on a sparsity-promoting bi-level optimization problem. Finally, we explore a meta-learning version of this algorithm based on group Lasso multiclass SVM base-predictors, for which we derive a tractable dual formulation. It obtains competitive results on standard few-shot classification benchmarks, while each task is using only a fraction of the learned representations.

## 1 Introduction

The recent literature on self-supervised learning has provided evidence that learning a representation on large corpuses of data can yield strong performances on a wide variety of downstream tasks (Devlin et al., 2018; Chen et al., 2020), especially in few-shot learning scenarios where the training data for these tasks is limited (Brown et al., 2020b; Dosovitskiy et al., 2021; Radford et al., 2021). Beyond transferring across multiple tasks, these learned representations also lead to improved robustness against distribution shifts (Wortsman et al., 2022) as well as stunning text-conditioned image generation (Ramesh et al., 2022). However, preliminary assessments of the latter has highlighted shortcomings related to compositionality (Marcus et al., 2022), suggesting new algorithmic innovations are needed to make further progress.

Another line of work has argued for the integration of ideas from causality to make progress towards more robust and transferable machine learning systems (Pearl, 2019; Schölkopf, 2019; Goyal & Bengio, 2022). *Causal representation learning* has emerged recently as a field aiming to define and learn representations suited for causal reasoning (Schölkopf et al., 2021). This set of ideas is strongly related to learning *disentangled representations* (Bengio et al., 2013). Informally, a representation is considered disentangled when its components are in one-to-one correspondence with natural and interpretable factors of variations, such as object positions, colors or shape. Although a plethora of works have investigated theoretically under which conditions disentanglement is possible (Hyvärinen & Morioka, 2016; 2017; Hyvärinen et al., 2019; Khemakhem et al., 2020a; Locatello et al., 2020a; Klindt et al., 2021; Von Kügelgen et al., 2021; Gresele et al., 2021; Lachapelle et al., 2022; Lippe et al., 2022b; Ahuja et al., 2022c), fewer works have tackled *how a disentangled representation could be beneficial for downstream tasks*. Those who did mainly provide empirical rather than theoretical evidence for or against its usefulness (Locatello et al., 2019; van Steenkiste et al., 2019; Miladinović et al., 2019; Dittadi et al., 2021; Montero et al., 2021).

In this work, we explore synergies between disentanglement and sparse base-predictors in the context of multi-task learning. At the heart of our contributions is the assumption that only a small subset of all factors of variations are useful for each downstream task, and this subset might change from one task to another. We will refer to such tasks as *sparse tasks*, and their corresponding sets of useful factors as their *supports*. This assumption was initially suggested by Bengio et al. (2013, Section 3.5): "the feature set being trained may be destined to be used in multiple tasks that may have distinct [and unknown] subsets of relevant features. Considerations such as these lead us to

the conclusion that the most robust approach to feature learning is to disentangle as many factors as possible, discarding as little information about the data as is practical". This strategy is very much in line with the current self-supervised learning trend (Radford et al., 2021), except for its focus on disentanglement.

Our main contributions are the following: (i) We formalize this "sparse task assumption" and argue theoretically and empirically how, in this context, disentangled representations coupled with sparsity-regularized base-predictors can obtain better generalization than their entangled counterparts (Section 2.1). (ii) We introduce a novel identifiability result (Theorem 1) which shows how one can leverage multiple sparse tasks to learn a shared disentangled representation by regularizing the task-specific predictors to be maximally sparse (Section 2.2.1). Crucially, Assumption 7 formalizes how diverse the task supports have to be in order to guarantee disentanglement. (iii) Motivated by this result, we propose a tractable bi-level optimization (Problem (4)) to learn the shared representation while regularizing the task-specific base-predictors to be sparse (Section 2.2.2). We validate our theory by showing our approach can indeed disentangle latent factors on tasks constructed from the 3D Shapes dataset (Burgess & Kim, 2018). (iv) Finally, we draw a connection between this bi-level optimization problem and some formulations from the meta-learning literature (Section 2.3). Inspired by our identifiability result, we enhance an existing method (Lee et al., 2019), where the base-learners are now group-sparse SVMs. We show that this new meta-learning algorithm achieves competitive performance on the *mini*ImageNet benchmark (Vinyals et al., 2016), while only using a fraction of the learned representation.

## 2    SYNERGIES BETWEEN DISENTANGLEMENT AND SPARSITY

In this section, we formally introduce the notion of entangled and disentangled representations. First, we assume the existence of some ground-truth encoder function $f_\theta : \mathbb{R}^d \to \mathbb{R}^m$ that maps observations $x \in \mathcal{X} \subseteq \mathbb{R}^d$, *e.g.*, images, to its corresponding interpretable and usually lower dimensional representation $f_\theta(x) \in \mathbb{R}^m$, $m \leq d$. The exact form of this ground-truth encoder depends on the task at hand, but also on what the machine learning practitioner considers as interpretable. The learned encoder function is denoted by $f_{\hat\theta} : \mathbb{R}^d \to \mathbb{R}^m$, and should not be conflated with the ground-truth representation $f_\theta$. For example, $f_{\hat\theta}$ can be parametrized by a neural network. Throughout, we are going to use the following definition of disentanglement.

**Definition 1** (Disentangled Representation, Khemakhem et al. 2020a; Lachapelle et al. 2022). *A learned encoder function $f_{\hat\theta} : \mathbb{R}^d \to \mathbb{R}^m$ is said to be* disentangled *w.r.t. the ground-truth representation $f_\theta$ when there exists an invertible diagonal matrix $D$ and a permutation matrix $P$ such that, for all $x \in \mathcal{X}$, $f_{\hat\theta}(x) = DP f_\theta(x)$. Otherwise the encoder $f_{\hat\theta}$ is said to be* entangled.

Intuitively, a representation is disentangled when there is a one-to-one correspondence between its components and the components of the ground-truth representation, up to rescaling. Note that there exist less stringent notions of disentanglement which allow for component-wise nonlinear invertible transformations of the factors (Hyvärinen & Morioka, 2017; Hyvärinen et al., 2019).

**Notation.** Capital bold letters denote matrices and lower case bold letters denote vectors. The set of integers from 1 to $n$ is denoted by $[n]$. We write $\|\cdot\|$ for the Euclidean norm on vectors and the Frobenius norm on matrices. For a matrix $A \in \mathbb{R}^{k \times m}$, $\|A\|_{2,1} = \sum_{j=1}^m \|A_{:j}\|$, and $\|A\|_{2,0} = \sum_{j=1}^m \mathbb{1}_{\|A_{:j}\| \neq 0}$, where $\mathbb{1}$ is the indicator function. The ground-truth parameter of the encoder function is $\theta$, while that of the learned representation is $\hat\theta$. We follow this convention for all the parameters throughout. Table 1 in Appendix A summarizes all the notation.

### 2.1    DISENTANGLEMENT AND SPARSE BASE-PREDICTORS FOR IMPROVED GENERALIZATION

In this section, we compare the generalization performance of entangled and disentangled representations on sparse downstream tasks. We show that the maximum likelihood estimator (defined in Problem (1)) computed on *linearly equivalent* representations (entangled or disentangled) yield the same model (Proposition 1). However, disentangled representations have better generalization properties when combined with a sparse base-predictor (Proposition 2 and Figure 1).

First, the learned representation $f_{\hat\theta}$ is assumed to be *linearly equivalent* to the ground-truth representation $f_\theta$, i.e. there exists an invertible matrix $L$ such that, for all $x \in \mathcal{X}$, $f_{\hat\theta}(x) = L f_\theta(x)$.

Note that despite being assumed linearly equivalent, the learned representation $\boldsymbol{f}_{\hat{\boldsymbol{\theta}}}$ might not be disentangled (Definition 1); in that case, we say the representation is *linearly entangled*. When we refer to a disentangled representation, we write $\boldsymbol{L} := \boldsymbol{DP}$. Roeder et al. (2021) have shown that many common methods learn representations identifiable up to linear equivalence, such as deep neural networks for classification, contrastive learning (Oord et al., 2018; Radford et al., 2021) and autoregressive language models (Mikolov et al., 2010; Brown et al., 2020a).

Consider the following maximum likelihood estimator (MLE):[1]

$$\hat{\boldsymbol{W}}_n^{(\hat{\boldsymbol{\theta}})} := \arg\max_{\tilde{\boldsymbol{W}}} \sum_{(\boldsymbol{x},y)\in\mathcal{D}} \log p(y; \boldsymbol{\eta} = \tilde{\boldsymbol{W}} \boldsymbol{f}_{\hat{\boldsymbol{\theta}}}(\boldsymbol{x})), \tag{1}$$

where $y$ denotes the label, $\mathcal{D} := \{(\boldsymbol{x}^{(i)}, y^{(i)})\}_{i=1}^n$ is the dataset, $p(y; \boldsymbol{\eta})$ is a distribution over labels parameterized by $\boldsymbol{\eta} \in \mathbb{R}^k$, and $\hat{\boldsymbol{W}} \in \mathbb{R}^{k\times m}$ is the task-specific predictor[2]. The following result shows that the maximum likelihood estimator defined in Problem (1) is invariant to invertible linear transformations of the features. Note that it is an almost direct consequence of the invariance of MLE to reparametrization (Casella & Berger, 2001, Thm. 7.2.10). See Appendix A for a proof.

**Proposition 1** (MLE Invariance to Invertible Linear Transformations of the Features)**.** *Let $\hat{\boldsymbol{W}}_n^{(\hat{\boldsymbol{\theta}})}$ and $\hat{\boldsymbol{W}}_n^{(\boldsymbol{\theta})}$ be the solutions to Problem (1) with the representations $\boldsymbol{f}_{\hat{\boldsymbol{\theta}}}$ and $\boldsymbol{f}_{\boldsymbol{\theta}}$, respectively (which we assume are unique). If there exists an invertible matrix $\boldsymbol{L}$ such that, $\forall \boldsymbol{x} \in \mathcal{X}$, $\boldsymbol{f}_{\hat{\boldsymbol{\theta}}}(\boldsymbol{x}) = \boldsymbol{L} \boldsymbol{f}_{\boldsymbol{\theta}}(\boldsymbol{x})$; then we have, $\forall \boldsymbol{x} \in \mathcal{X}$, $\hat{\boldsymbol{W}}_n^{(\hat{\boldsymbol{\theta}})} \boldsymbol{f}_{\hat{\boldsymbol{\theta}}}(\boldsymbol{x}) = \hat{\boldsymbol{W}}_n^{(\boldsymbol{\theta})} \boldsymbol{f}_{\boldsymbol{\theta}}(\boldsymbol{x})$.*

Proposition 1 shows that the model $p(y; \hat{\boldsymbol{W}}_n^{(\hat{\boldsymbol{\theta}})} \boldsymbol{f}_{\hat{\boldsymbol{\theta}}}(\boldsymbol{x}))$ learned by Problem (1) is independent of $\boldsymbol{L}$, *i.e.,* the model is the same for disentangled and linearly entangled representations. We thus expect both disentangled and linearly entangled representations to perform identically on downstream tasks.

In what follows, we assume the data is generated according to the following process.

**Assumption 1** (Data generation process)**.** *The input-label pairs are i.i.d. samples from the distribution $p(\boldsymbol{x}, y) := p(y \mid \boldsymbol{x}) p(\boldsymbol{x})$ with $p(y \mid \boldsymbol{x}) := p(y; \boldsymbol{W} \boldsymbol{f}_{\boldsymbol{\theta}}(\boldsymbol{x}))$, where $\boldsymbol{W} \in \mathbb{R}^{k\times m}$ is the ground-truth coefficient matrix.*

To formalize the hypothesis that *only a subset of the features $\boldsymbol{f}_{\boldsymbol{\theta}}(\boldsymbol{x})$ are actually useful to predict the target $y$*, we assume that the ground-truth coefficient matrix $\boldsymbol{W}$ is column sparse, *i.e.,* $\|\hat{\boldsymbol{W}}\|_{2,0} = \ell < m$. Under this assumption, it is natural to constrain the MLE as such:

$$\hat{\boldsymbol{W}}_n^{(\hat{\boldsymbol{\theta}},\ell)} := \arg\max_{\tilde{\boldsymbol{W}}} \sum_{(\boldsymbol{x},y)\in\mathcal{D}} \log p(y; \tilde{\boldsymbol{W}} \boldsymbol{f}_{\hat{\boldsymbol{\theta}}}(\boldsymbol{x})) \quad \text{s.t.} \quad \|\tilde{\boldsymbol{W}}\|_{2,0} \le \ell. \tag{2}$$

The following proposition will help us understand how this additional constraint interacts with representations that are disentangled or linearly entangled. See Appendix A for a proof.

**Proposition 2** (Population MLE for Linearly Entangled Representations)**.** *Let $\hat{\boldsymbol{W}}_\infty^{(\hat{\boldsymbol{\theta}})}$ be the solution of the population-based MLE, $\arg\max_{\tilde{\boldsymbol{W}}} \mathbb{E}_{p(\boldsymbol{x},y)} \log p(y; \tilde{\boldsymbol{W}} \boldsymbol{f}_{\hat{\boldsymbol{\theta}}}(\boldsymbol{x}))$ (assumed to be unique). Suppose $\boldsymbol{f}_{\hat{\boldsymbol{\theta}}}$ is linearly equivalent to $\boldsymbol{f}_{\boldsymbol{\theta}}$, and Assumption 1 holds, then, $\hat{\boldsymbol{W}}_\infty^{(\hat{\boldsymbol{\theta}})} = \boldsymbol{W} \boldsymbol{L}^{-1}$.*

From Proposition 2, one can see that if the representation $\boldsymbol{f}_{\hat{\boldsymbol{\theta}}}$ is disentangled, then $\|\hat{\boldsymbol{W}}_\infty^{(\hat{\boldsymbol{\theta}})}\|_{2,0} = \|\boldsymbol{W}(\boldsymbol{DP})^{-1}\|_{2,0} = \|\boldsymbol{W}\|_{2,0} = \ell$. Thus, in that case, the sparsity constraint in Problem (2) does not exclude the population MLE estimator from its hypothesis class, and yields a decrease in the generalization gap (Bickel et al., 2009; Lounici et al., 2011a; Mohri et al., 2018) without biasing the estimator. Contrarily, when $\boldsymbol{f}_{\hat{\boldsymbol{\theta}}}$ is linearly entangled, the population MLE might have more nonzero columns than the ground-truth, and thus would be excluded from the hypothesis space of Problem (2), which, in turn, would bias the estimator.

**Empirical validation.** We now present a simple simulated experiment to validate the above claim that *disentangled representations coupled with sparsity regularization can have better generalization.* Figure 1 compares the generalization performance of the convex relaxation of Problem (2)

---

[1]We assume the solution is unique.

[2]$p(y; \boldsymbol{\eta})$ could be a Gaussian density (regression) or a categorical distribution (classification).

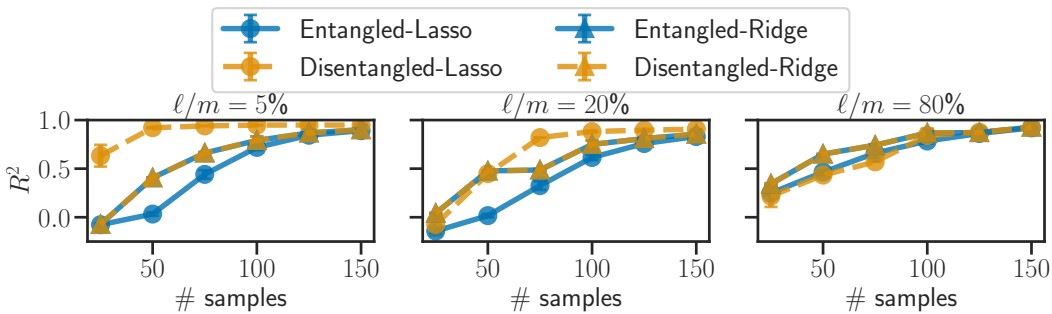

Figure 1: Test performance for the entangled and disentangled representation using Lasso and Ridge regression. All the results are averaged over 10 seeds, with standard error shown in error bars.

(Lasso regression, Tibshirani 1996) and Ridge regression (Hoerl & Kennard, 1970) on both disentangled and linearly entangled representations. Lasso regression coupled with the disentangled representation obtains better generalization than the other alternatives when $\ell/m = 5\%$ and when the number of samples is very small. We can also see that, disentanglement, sparsity regularization and sufficient sparsity in the ground-truth data generating process are necessary to see a significant improvement, in line with our discussion. Lastly, the performance of all methods converge to the same value when the number of samples grows. See Appendix D.1 for more details and discussion on the results.

## 2.2 DISENTANGLEMENT VIA SPARSE MULTITASK LEARNING

In Section 2.1, we argued that disentangled representations can improve generalization when combined with sparse base-predictors, but we did not provide an approach to learn them. We first provide a new identification result (Theorem 1, Section 2.2.1), which states that in the context of sparse multitask learning, sparse base-predictors yield disentangled representations. Then, in Section 2.2.2, we provide a practical way to learn disentangled representations motivated by our identifiability result.

Throughout this section, we assume the learner is given a set of $T$ datasets $\{\mathcal{D}_1, \ldots, \mathcal{D}_T\}$ where each dataset $\mathcal{D}_t := \{(\boldsymbol{x}^{(t,i)}, y^{(t,i)})\}_{i=1}^n$ consists of $n$ couples of input $\boldsymbol{x} \in \mathbb{R}^d$ and label $y \in \mathcal{Y}$. The set of labels $\mathcal{Y}$ might contain either class indices or real values, depending on whether we are concerned with classification or regression tasks.

### 2.2.1 IDENTIFIABILITY ANALYSIS

We now present the main theoretical result of our work which shows how learning a shared representation across tasks while penalizing the task-specific base-predictor to be sparse can induce disentanglement. Our theory relies on the following ground-truth data generating process:

**Assumption 2** (Ground-truth data generating process). *For each task $t$, the dataset $\mathcal{D}_t$ is made of i.i.d. samples from the distribution $p(\boldsymbol{x}, y \mid \boldsymbol{W}^{(t)}) := p(y \mid \boldsymbol{x}, \boldsymbol{W}^{(t)})p(\boldsymbol{x} \mid \boldsymbol{W}^{(t)})$ with $p(y \mid \boldsymbol{x}, \boldsymbol{W}^{(t)}) := p(y; \boldsymbol{W}^{(t)} \boldsymbol{f_\theta}(\boldsymbol{x}))$, where $\boldsymbol{W}^{(t)} \in \mathbb{R}^{k \times m}$ is the task-specific ground-truth coefficient matrix. Moreover, the matrices $\boldsymbol{W}^{(t)}$ are i.i.d. samples from some probability measure $\mathbb{P}_{\boldsymbol{W}}$ with support $\mathcal{W}$. Also, for all $\boldsymbol{W} \in \mathcal{W}$, the support of $p(\boldsymbol{x} \mid \boldsymbol{W})$ is $\mathcal{X} \subseteq \mathbb{R}^d$ (fixed across tasks).*

The above assumption states that (i) the ground-truth coefficient matrices $\boldsymbol{W}^{(t)}$ are task-specific while the representation $\boldsymbol{f_\theta}$ is shared across all the tasks, (ii) the task-specific $\boldsymbol{W}^{(t)}$ are sampled i.i.d. from some distribution $\mathbb{P}_{\boldsymbol{W}}$, and (iii) the support of $\boldsymbol{x}$ is shared across tasks.

**Assumption 3** (Identifiability of $\boldsymbol{\eta}$). *The parameter $\boldsymbol{\eta}$ is identifiable from $p(y; \boldsymbol{\eta})$, i.e. $\forall y; \ p(y; \boldsymbol{\eta}) = p(y; \tilde{\boldsymbol{\eta}}) \implies \boldsymbol{\eta} = \tilde{\boldsymbol{\eta}}$.*

This property holds, *e.g.*, when $p(y; \boldsymbol{\eta})$ is a Gaussian in the usual $\mu, \sigma^2$ parameterization. Generally, it also holds for minimal parameterizations of exponential families (Wainwright & Jordan, 2008).

The following assumption requires the ground-truth representation $\boldsymbol{f_\theta}(\boldsymbol{x})$ to vary enough such that its image cannot be trapped inside a proper subspace.

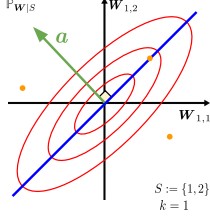

Figure 2: Illustration of Assumption 6 showing three examples of distribution $\mathbb{P}_{\boldsymbol{W}|S}$. The red distribution satisfies the assumption, but the blue and orange distributions do not. The red lines are level sets of a Gaussian distribution with full rank covariance. The blue line represents the support of a Gaussian distribution with a low rank covariance. The orange dots represents a distribution with finite support. The green vector $\boldsymbol{a}$ shows that the condition is violated for both the blue and the orange distribution, since, in both cases, $\boldsymbol{W}_{1,S}\boldsymbol{a} = 0$ (orthogonal) with probability greater than zero.

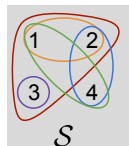 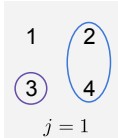 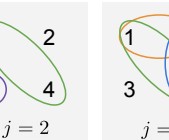 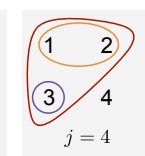

Figure 3: The leftmost figure represents $\mathcal{S}$, the support of some $p(S)$. The other figures form a verification that Assumption 7 holds for $\mathcal{S}$.

**Assumption 4** (Sufficient representation variability). *There exists $\boldsymbol{x}^{(1)}, \dots, \boldsymbol{x}^{(m)} \in \mathcal{X}$ such that the matrix $\boldsymbol{F} := [\boldsymbol{f}_{\boldsymbol{\theta}}(\boldsymbol{x}^{(1)}), \dots, \boldsymbol{f}_{\boldsymbol{\theta}}(\boldsymbol{x}^{(m)})]$ is invertible.*

The following assumption requires that the support of the distribution $\mathbb{P}_{\boldsymbol{W}}$ is sufficiently rich.

**Assumption 5** (Sufficient task variability). *There exists $\boldsymbol{W}^{(1)}, \dots, \boldsymbol{W}^{(m)} \in \mathcal{W}$ and row indices $i_1, \dots, i_m \in [k]$ such that the rows $\boldsymbol{W}^{(1)}_{i_1,:}, \dots, \boldsymbol{W}^{(m)}_{i_m,:}$ are linearly independent.*

Under Assumptions 2 to 5 the representation $\boldsymbol{f}_{\boldsymbol{\theta}}$ is identifiable up to linear equivalence (see Theorem 2 in Appendix B). Similar results where shown by Roeder et al. (2021); Ahuja et al. (2022c). The next assumptions will guarantee disentanglement.

In order to formalize the intuitive idea that most tasks do not require all features, we will denote by $S^{(t)}$ the support of the matrix $\boldsymbol{W}^{(t)}$, i.e. $S^{(t)} := \{j \in [m] \mid \boldsymbol{W}^{(t)}_{:,j} \neq \boldsymbol{0}\}$. In other words, $S^{(t)}$ is the set of features which are useful to predict $y$ in the $t$-th task; note that it is unknown to the learner. For our analysis, we decompose $\mathbb{P}_{\boldsymbol{W}}$ as $\mathbb{P}_{\boldsymbol{W}} = \sum_{S \in \mathcal{P}([m])} p(S)\mathbb{P}_{\boldsymbol{W}|S}$, where $\mathcal{P}([m])$ is the collection of all subsets of $[m]$, $p(S)$ is the probability that the support of $\boldsymbol{W}$ is $S$ and $\mathbb{P}_{\boldsymbol{W}|S}$ is the conditional distribution of $\boldsymbol{W}$ given that its support is $S$. Let $\mathcal{S}$ be the support of the distribution $p(S)$, i.e. $\mathcal{S} := \{S \in \mathcal{P}([m]) \mid p(S) > 0\}$. The set $\mathcal{S}$ will have an important role in Assumption 7 & Theorem 1.

The following assumption requires that $\mathbb{P}_{\boldsymbol{W}|S}$ does not concentrate on certain proper subspaces.

**Assumption 6** (Intra-support sufficient task variability). *For all $S \in \mathcal{S}$ and all $\boldsymbol{a} \in \mathbb{R}^{|S|}\backslash 0$, $\mathbb{P}_{\boldsymbol{W}|S}\{\boldsymbol{W} \in \mathbb{R}^{k \times m} \mid \boldsymbol{W}_{:S}\boldsymbol{a} = \boldsymbol{0}\} = 0$.*

We illustrate the above assumption in the simpler case where $k = 1$. For instance, Assumption 6 holds when the distribution of $\boldsymbol{W}_{1,S} \mid S$ has a density w.r.t. the Lebesgue measure on $\mathbb{R}^{|S|}$, which is true for example when $\boldsymbol{W}_{1,S} \mid S \sim \mathcal{N}(\boldsymbol{0}, \boldsymbol{\Sigma})$ and the covariance matrix $\boldsymbol{\Sigma}$ is full rank (red distribution in Figure 2). However, if $\boldsymbol{\Sigma}$ is not full rank, the probability distribution of $\boldsymbol{W}_{1,S} \mid S$ concentrates its mass on a proper linear subspace $V \subsetneq \mathbb{R}^{|S|}$, which violates Assumption 6 (blue distribution in Figure 2). Another important counter-example is when $\mathbb{P}_{\boldsymbol{W}|S}$ concentrates some of its mass on a point $\boldsymbol{W}^{(0)}$, i.e. $\mathbb{P}_{\boldsymbol{W}|S}\{\boldsymbol{W}^{(0)}\} > 0$ (orange distribution in Figure 2). Interestingly, there are distributions over $\boldsymbol{W}_{1,S} \mid S$ that do not have a density w.r.t. the Lebesgue measure, but still satisfy Assumption 6. This is the case, e.g., when $\boldsymbol{W}_{1,S} \mid S$ puts uniform mass over a $(|S|-1)$-dimensional sphere embedded in $\mathbb{R}^{|S|}$ and centered at zero. See Appendix B.2 for a justification.

The following assumption requires that the support $\mathcal{S}$ of $p(S)$ is "rich enough".

**Assumption 7** (Sufficient support variability). *For all $j \in [m]$, $\bigcup_{S \in \mathcal{S} \mid j \notin S} S = [m] \setminus \{j\}$.*

Intuitively, Assumption 7 requires that, for every feature $j$, one can find a set of tasks such that their supports cover all features except $j$ itself. Figure 3 shows an example of $\mathcal{S}$ satisfying Assumption 7. Removing the latter would only yield *partial disentanglement* (Lachapelle & Lacoste-Julien, 2022).

We are now ready to show the main theoretical result of this work, which provides a bi-level optimization problem for which the optimal representations are guaranteed to be disentangled. It as-

sumes infinitely many tasks are observed, with task-specific ground-truth matrices $\boldsymbol{W}$ sampled from $\mathbb{P}_{\boldsymbol{W}}$. We denote by $\hat{\boldsymbol{W}}^{(\boldsymbol{W})}$ the task-specific estimator of $\boldsymbol{W}$. See Appendix B.1 for a proof. Note that we suggest a tractable relaxation in Section 2.2.2.

**Theorem 1** (Sparse multi-task learning for disentanglement). *Let $\hat{\boldsymbol{\theta}}$ be a minimizer of*

$$\min_{\hat{\boldsymbol{\theta}}} \mathbb{E}_{\mathbb{P}_{\boldsymbol{W}}} \mathbb{E}_{p(\boldsymbol{x},y|\boldsymbol{W})} - \log p(y; \hat{\boldsymbol{W}}^{(\boldsymbol{W})} \boldsymbol{f}_{\hat{\boldsymbol{\theta}}}(\boldsymbol{x}))$$

$$\text{s.t.} \quad \forall \boldsymbol{W} \in \mathcal{W}, \ \hat{\boldsymbol{W}}^{(\boldsymbol{W})} \in \underset{\substack{\tilde{\boldsymbol{W}} \ \text{s.t.} \\ ||\tilde{\boldsymbol{W}}||_{2,0} \leq ||\boldsymbol{W}||_{2,0}}}{\arg\min} \mathbb{E}_{p(\boldsymbol{x},y|\boldsymbol{W})} - \log p(y; \tilde{\boldsymbol{W}} \boldsymbol{f}_{\hat{\boldsymbol{\theta}}}(\boldsymbol{x})) \ . \tag{3}$$

*Then, under Assumptions 2 to 7, $\boldsymbol{f}_{\hat{\boldsymbol{\theta}}}$ is disentangled w.r.t. $\boldsymbol{f}_{\boldsymbol{\theta}}$ (Definition 1).*

Intuitively, this optimization problem effectively selects a representation $\boldsymbol{f}_{\hat{\boldsymbol{\theta}}}$ that (i) allows a perfect fit of the data distribution, and (ii) allows the task-specific estimators $\hat{\boldsymbol{W}}^{(\boldsymbol{W})}$ to be as sparse as the ground-truth $\boldsymbol{W}$. With the same disentanglement guarantees, Theorem 4 in Appendix B presents a variation of Problem (3) which enforces the weaker constraint $\mathbb{E}_{\mathbb{P}_{\boldsymbol{W}}} \|\hat{\boldsymbol{W}}^{(\boldsymbol{W})}\|_{2,0} \leq \mathbb{E}_{\mathbb{P}_{\boldsymbol{W}}} \|\boldsymbol{W}\|_{2,0}$, instead of $\|\hat{\boldsymbol{W}}^{(\boldsymbol{W})}\|_{2,0} \leq \|\boldsymbol{W}\|_{2,0}$ for each task $\boldsymbol{W}$ individually.

### 2.2.2 Tractable bilevel optimization problems for sparse multitask learning

Problem (3) was shown to yield a disentangled representation (Theorem 1), but is intractable due to the $L_{2,0}$-seminorm. Thus we use the $L_{2,1}$ convex relaxation of the $L_{2,0}$-seminorm, which is also known to promote group sparsity (Obozinski et al., 2006; Argyriou et al., 2008; Lounici et al., 2009):

$$\min_{\hat{\boldsymbol{\theta}}} \ -\frac{1}{Tn} \sum_{t=1}^{T} \sum_{(\boldsymbol{x},y) \in \mathcal{D}_t} \log p(y; \hat{\boldsymbol{W}}^{(t)} \boldsymbol{f}_{\hat{\boldsymbol{\theta}}}(\boldsymbol{x}))$$

$$\text{s.t.} \quad \forall t \in [T], \ \hat{\boldsymbol{W}}^{(t)} \in \arg\min_{\tilde{\boldsymbol{W}}} -\frac{1}{n} \sum_{(\boldsymbol{x},y) \in \mathcal{D}_t} \log p(y; \tilde{\boldsymbol{W}} \boldsymbol{f}_{\hat{\boldsymbol{\theta}}}(\boldsymbol{x})) + \lambda_t ||\tilde{\boldsymbol{W}}||_{2,1} \ . \tag{4}$$

Following Bengio (2000); Pedregosa (2016), one can compute the (hyper)gradient of the outer function using implicit differentiation, even if the inner optimization problem is non-smooth (Bertrand et al., 2020; Bolte et al., 2021; Malézieux et al., 2022; Bolte et al., 2022). Once the hypergradient is computed, one can optimize Problem (4) using usual first-order methods (Wright & Nocedal, 1999).

Note that the quantity $\hat{\boldsymbol{W}}^{(t)} \boldsymbol{f}_{\hat{\boldsymbol{\theta}}}(\boldsymbol{x})$ is invariant to simultaneous rescaling of $\hat{\boldsymbol{W}}^{(t)}$ by a scalar and of $\boldsymbol{f}_{\hat{\boldsymbol{\theta}}}(\boldsymbol{x})$ by its inverse. Thus, without constraints on $\boldsymbol{f}_{\hat{\boldsymbol{\theta}}}(\boldsymbol{x})$, $\|\hat{\boldsymbol{W}}^{(t)}\|_{2,1}$ can be made arbitrarily small. This is a usual problem in sparse dictionary learning (Kreutz-Delgado et al., 2003; Mairal et al., 2008; 2009; 2011), where unit-norm constraints are usually imposed on the column of the dictionary. Here, since $\boldsymbol{f}_{\hat{\boldsymbol{\theta}}}$ is parametrized by a neural network, we suggest to apply batch or layer normalization (Ioffe & Szegedy, 2015; Ba et al., 2016) to control its norm. Since the number of relevant features might be task-dependent, Problem (4) has one regularization hyperparameter $\lambda_t$ per task. To limit the number of hyperparameters in practice, we select $\lambda_t := \lambda$ for all $t \in [T]$.

### 2.3 Link with meta-learning

In the setting known as meta-learning (Finn et al., 2017), for a large number of tasks $T$, we are given training datasets $\mathcal{D}_t^{\text{train}}$, which usually contains a small number of samples $n$. Unlike in the multi-task setting though (*i.e.*, unlike in Section 2.2), we are also given separated test datasets $\mathcal{D}_t^{\text{test}}$ to evaluate how well the learned model generalizes to new test samples. *In meta-learning, the goal is to learn a training procedure which will generalize well on out-of-distribution tasks.* The bi-level formulation Problem (4) is closely related to *metric-based meta-learning* (Snell et al., 2017; Bertinetto et al., 2019), where a shared representation $\boldsymbol{f}_{\hat{\boldsymbol{\theta}}}$ is learned across all tasks. The representation is jointly learned with simple task-specific classifiers, which are usually optimization-based classifiers, such as support-vector machines. Formally, *metric-based meta-learning* can be formulated as follows

$$\min_{\hat{\boldsymbol{\theta}}} \sum_{t=1}^{T} \sum_{(\boldsymbol{x},y) \in \mathcal{D}_t^{\text{test}}} \mathcal{L}_{\text{out}}(\hat{\boldsymbol{W}}_{\hat{\boldsymbol{\theta}}}^{(t)}; f_{\hat{\boldsymbol{\theta}}}(\boldsymbol{x}), y) \qquad \text{s.t.} \ \hat{\boldsymbol{W}}_{\hat{\boldsymbol{\theta}}}^{(t)} \in \arg\min_{\tilde{\boldsymbol{W}}} \sum_{(\boldsymbol{x},y) \in \mathcal{D}_t^{\text{train}}} \mathcal{L}_{\text{in}}(\tilde{\boldsymbol{W}}; f_{\hat{\boldsymbol{\theta}}}(\boldsymbol{x}), y).$$

Inspired by Lee et al. (2019), where the base-classifiers were multiclass support-vector machines (SVMs, Crammer & Singer 2001), we propose to use group Lasso penalized multiclass SVMs, in order to introduce sparsity in the base-learners, with $\boldsymbol{Y} \in \mathbb{R}^{n \times k}$ the one-hot encoding of $\boldsymbol{y} \in \mathbb{R}^n$:

$$\mathcal{L}_{\text{in}}(\boldsymbol{W}; f_{\hat{\boldsymbol{\theta}}}(\boldsymbol{x})), y) := \max_{l \in [k]} \left( (\boldsymbol{W}_{\boldsymbol{y}_i:} - \boldsymbol{W}_{l:}) \cdot f_{\hat{\boldsymbol{\theta}}}(\boldsymbol{x}) - \boldsymbol{Y}_{:l} \right) + \tfrac{\lambda_1}{n}\|\boldsymbol{W}\|_{2,1} + \tfrac{\lambda_2}{2n}\|\boldsymbol{W}\|^2. \quad (5)$$

In few-shot learning settings, the number of features $m$ is usually much larger than the number of samples $n$ (in Lee et al. 2019, $m = 1.6\cdot 10^4$ and $n \leq 25$). In such scenarios, SVMs-like problems are usually solved through their dual (Boyd et al., 2004, Chap. 5) problems, for computational (Hsieh et al., 2008) and theoretical (Shalev-Shwartz & Zhang, 2012) benefits.

**Proposition 3.** (Dual Group Lasso Soft-Margin Multiclass SVM.) *The dual of the inner problem with $\mathcal{L}_{\text{in}}$ as defined in (5) writes*

$$\min_{\boldsymbol{\Lambda} \in \mathbb{R}^{n \times k}} \frac{1}{\lambda_2} \sum_{j=1}^{m} \|\text{BST}\left((\boldsymbol{Y} - \boldsymbol{\Lambda})^\top \boldsymbol{F}_{:j}, \lambda_1\right)\|^2 + \langle \boldsymbol{Y}, \boldsymbol{\Lambda} \rangle + \sum_{i=1}^{n} \mathbb{1}_{\sum_{l=1}^{k} \boldsymbol{\Lambda}_{il} = 1} + \sum_{i=1}^{n}\sum_{l=1}^{k} \mathbb{1}_{\boldsymbol{\Lambda}_{il} \geq 0}, \quad (6)$$

*with* BST *the block soft-thresholding operator:* $\text{BST} : (\boldsymbol{a}, \tau) \mapsto (1 - \tau/\|\boldsymbol{a}\|)_+ \boldsymbol{a}$, $\boldsymbol{F} \in \mathbb{R}^{n \times m}$ *the concatenation of* $\{f_{\hat{\boldsymbol{\theta}}}(x)\}_{(x,y) \in \mathcal{D}^{\text{train}}}$. *In addition, the primal-dual link writes, for all $j \in [m]$,* $\boldsymbol{W}_{:j} = \text{BST}\left((\boldsymbol{Y} - \boldsymbol{\Lambda})^\top \boldsymbol{F}_{:j}, \lambda_1\right)/\lambda_2$.

Proof of Proposition 3 can be found in Appendix C.1. The objective of Problem (6) is composed of a smooth term and block separable non-smooth term, hence it can be solved efficiently using proximal block coordinate descent (Tseng, 2001). As stated in Section 2.2, argmin differentiation of the solution of Problem (6) can be done using implicit differentiation (Bertrand et al., 2022). Although Theorem 1 is not directly applicable to the meta-learning formulation proposed in this section, we conjecture that similar techniques could be reused to prove an identifiability result in this setting.

## 3 RELATED WORK

**Disentanglement.** Since the work of Bengio et al. (2013), many methods have been proposed to learn disentangled representations based on various heuristics (Higgins et al., 2017; Chen et al., 2018; Kim & Mnih, 2018; Kumar et al., 2018; Bouchacourt et al., 2018). Following the work of Locatello et al. (2019), which highlighted the lack of identifiability in modern deep generative models, many works have proposed more or less weak forms of supervision motivated by identifiability analyses (Locatello et al., 2020a; Klindt et al., 2021; Von Kügelgen et al., 2021; Ahuja et al., 2022a;c; Zheng et al., 2022). A similar line of work have adopted the causal representation learning perspective (Lachapelle et al., 2022; Lachapelle & Lacoste-Julien, 2022; Lippe et al., 2022b;a; Ahuja et al., 2022b; Yao et al., 2022; Brehmer et al., 2022). The problem of identifiability was well known among the *independent component analysis* (ICA) community (Hyvärinen et al., 2001; Hyvärinen & Pajunen, 1999) which came up with solutions for general nonlinear mixing functions by leveraging auxiliary information (Hyvärinen & Morioka, 2016; 2017; Hyvärinen et al., 2019; Khemakhem et al., 2020a;b). Another approach is to consider restricted hypothesis classes of mixing functions (Taleb & Jutten, 1999; Gresele et al., 2021). Contrarily to most of the above works, we do not assume that the inputs $\boldsymbol{x}$ are generated by transforming a latent random variable $\boldsymbol{z}$ through a bijective decoder $\boldsymbol{g}$. Instead, we assume the existence of a not necessarily bijective ground-truth feature extractor $\boldsymbol{f}_{\boldsymbol{\theta}}(\boldsymbol{x})$ from which the labels can be predicted using only a subset of its components in every tasks (Assumption 2). Many of these works make assumptions about the distribution of latent factors, e.g., (conditional) independence, exponential family or other parametric assumptions. In contrast, we make comparatively weaker assumptions on the support of the ground-truth features (Assumption 4), which are allowed to present dependencies (Section 4). Locatello et al. (2020b) proposed a semi-supervised learning approach to disentangle in cases where a few samples are labelled with the factors of variations themselves. This is different from our approach as the labels that we consider can be sampled from some $p(y; \boldsymbol{W} f_{\hat{\boldsymbol{\theta}}}(\boldsymbol{x}))$, which is more general. Ahuja et al. (2022c) consider a setting similar to ours, but they rely on the independence and non-gaussianity of the latent factors for disentanglement using linear ICA.

**Multi-task, transfer & invariant learning.** The statistical advantages of multi-task representation learning is well understood (Lounici et al., 2011a;b; Maurer et al., 2016). However, apart from Zhang

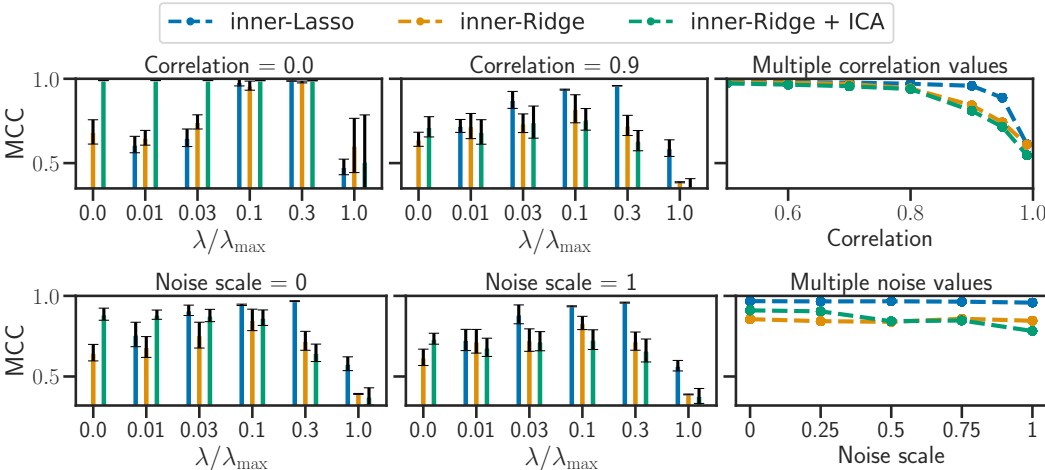

Figure 4: Disentanglement performance (MCC) for inner-Lasso, inner-Ridge and inner-Ridge combined with ICA as a function of the regularization parameter (left and middle). Varying level of correlation between latents (top) and of noise on the latents (bottom). The right columns shows performance of the best hyperparameter for different values of correlation and noise.

et al. (2022), theoretical benefits of disentanglement for transfer learning are not clearly established. Some works have investigated this question empirically and obtained both positive (van Steenkiste et al., 2019; Miladinović et al., 2019; Dittadi et al., 2021) and negative results (Locatello et al., 2019; Montero et al., 2021). Invariant risk minimization (Arjovsky et al., 2020; Ahuja et al., 2020; Krueger et al., 2021; Lu et al., 2021) aims at learning a representation that elicits a task-invariant base-predictor. This differs from our approach which learns base-predictors that are task-specific .

**Dictionary learning and sparse coding.** We contrast our approach, which jointly learns a *dense representation* and sparse base-predictors (Problem (4)), with the line of work which consists in learning *sparse representations* (Chen et al., 1998; Gribonval & Lesage, 2006). For instance, sparse dictionary learning (Mairal et al., 2009; 2011; Maurer et al., 2013) is an unsupervised technique which aims at learning sparse representations that refer to atoms of a learned dictionary. Contrarily to our method which computes the representation of a single input $x$ by evaluating a function approximator $f_{\hat{\theta}}$, in sparse dictionary learning, the representation of a single input is computed by minimizing a reconstruction loss. In the case of supervised dictionary learning (Mairal et al., 2008), an additional (potentially expressive) classifier is learned. This large literature has lead to a wide variety of estimators: for instance, Mairal et al. (2008, Eq. 4), which minimizes the sum of the classification error and the approximation error of the code, or Mairal et al. (2011); Malézieux et al. (2022), which introduce bi-level formulations which shares similarities with our formulations.

## 4 EXPERIMENTS

**Semi-real experiments on 3D Shapes.** We now illustrate Theorem 1 by applying Problem (4) to tasks generated using the 3D Shapes dataset (Burgess & Kim, 2018).

*Data generation.* For all tasks $t$, the labelled dataset $\mathcal{D}_t = \{(\boldsymbol{x}^{(t,i)}, y^{(t,i)})\}_{i=1}^n$ is generated by first sampling the ground-truth latent variables $\boldsymbol{z}^{(t,i)} := \boldsymbol{f}_{\boldsymbol{\theta}}(\boldsymbol{x}^{(t,i)})$ i.i.d. according to some distribution $p(\boldsymbol{z})$, while the corresponding input is obtained doing $\boldsymbol{x}^{(t,i)} := \boldsymbol{f}_{\boldsymbol{\theta}}^{-1}(\boldsymbol{z}^{(t,i)})$ ($\boldsymbol{f}_{\boldsymbol{\theta}}$ is invertible in 3D Shapes). Then, a sparse weight vector $\boldsymbol{w}^{(t)}$ is sampled randomly to compute the labels of each example as $y^{(t,i)} := \boldsymbol{w}^{(t)} \cdot \boldsymbol{x}^{(t,i)} + \epsilon^{(t,i)}$, where $\epsilon^{(t,i)}$ is independent Gaussian noise. Figure 4 explores various choice of $p(\boldsymbol{z})$, i.e. by varying the level of correlation between the latent variables and by varying the level of noise on the ground-truth latents. See Appendix D.2 for more details about the data generating process.

*Algorithms.* In this setting where $p(y; \eta)$ is a Gaussian with fixed variance, the inner problem of Problem (4) amounts to Lasso regression, we thus refer to this approach as inner-Lasso. We also evaluate a simple variation of Problem (4) in which the $L_1$ norm is replaced by an $L_2$ norm, and refer to it as inner-Ridge. In addition we evaluate the representation obtained by performing linear

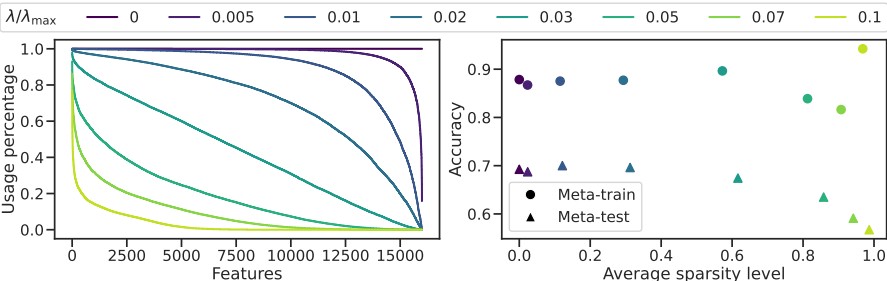

Figure 5: Effect of sparsity on the percentage of tasks using specific features, with our meta-learning objective, on *mini*ImageNet (left). The accuracy of the meta-learning algorithm and the average level of sparsity in the base-learners, as $\lambda$ varies (right).

ICA (Comon, 1992) on the representation learned by inner-Ridge: the case $\lambda = 0$ corresponds to the approach of Ahuja et al. (2022c).

*Discussion.* Figure 4 reports disentanglement performance of the three methods, as measured by the *mean correlation coefficient*, or MCC (Hyvärinen & Morioka, 2016; Khemakhem et al., 2020a) (Appendix D.2). In all settings, inner-Lasso obtains high MCC for some values of $\lambda$, being on par or surpassing the baselines. As the theory suggests, it is robust to high levels of correlations between the latents, as opposed to inner-Ridge with ICA which is very much affected by strong correlations (since ICA assumes independence). We can also see how additional noise on the latent variables hurts inner-Ridge with ICA while leaving inner-Lasso unaffected. Figure 6 in Appendix D.2 shows that all methods find a representation which is linearly equivalent to the ground-truth representation, except for very large values of $\lambda$. Refer to Appendix D.2 for more details. Appendix D.2.4 presents experiments showing to what extent inner-Lasso is robust to violations of Assumption 7. Appendix D.2.5 presents a visual evaluation of disentanglement. Appendix D.2.6 shows results for the same experiments with the DCI metric (Eastwood & Williams, 2018).

**Few-shot learning experiments.** Despite the lack of ground-truth latent factors in standard few-shot learning benchmarks, we also evaluate our meta-learning objective introduced in Section 2.3, using the dual formulation of the group Lasso penalized SVM as our base-learner, on the *mini*ImageNet dataset (Vinyals et al., 2016). The objective of this experiment is to show that the sparse formulation of the meta-learning objective is capable of reaching similar levels of performance, while using a fraction of the features. Details about the experimental settings are provided in Appendix D.3.

*Discussion.* In Figure 5 (left), we report how frequently the learned features are used by the base-learner on meta-training tasks; the gradual decrease in usage suggests that the features are reused in different contexts, across different tasks. We also observe (Figure 5, right) that adding sparsity to the base learner may also improve performance on meta-training tasks, while only using a fraction of all the features available in the learned representation, supporting our observations in Section 2.1 on the effect of sparsity on generalization on natural images (see Appendix D.3 for further discussion about how this still tests generalization). We also observe that some level of sparsity improves the performance on novel meta-test tasks, albeit to a smaller extent.

## 5 CONCLUSION

In this work, we investigated the synergies between sparsity, disentanglement and generalization. We showed that when the downstream task can be solved using only a fraction of the factors of variations, disentangled representations combined with sparse base-predictors can improve generalization (Section 2.1). Our novel identifiability result (Theorem 1) sheds light on how, in a multi-task setting, sparsity regularization on the task-specific predictors can induce disentanglement. This led to a practical bi-level optimization problem that was shown to yield disentangled representations on regression tasks based on the 3D Shapes dataset. Finally, we explored a meta-learning formulation extending this approach, and showed how sparse base-learners can help with generalization, while only using a small fraction of the features.

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

## CONTENTS

Table 1: Table of Notation.

### Norms & pseudonorms

| | | |
|---|---|---|
| $\|\cdot\|$ | | Euclidean norm on vectors and Frobenius norm on matrices |
| $\|\boldsymbol{A}\|_{2,1}$ | $:=$ | $\sum_{j=1}^{m} \|\boldsymbol{A}_{:j}\|$ |
| $\|\boldsymbol{A}\|_{2,0}$ | $:=$ | $\sum_{j=1}^{m} \mathbb{1}_{\|\boldsymbol{A}_{:j}\| \neq 0}$, where $\mathbb{1}$ is the indicator function. |

### Data

| | |
|---|---|
| $\boldsymbol{x} \in \mathbb{R}^d$ | Observations |
| $\mathcal{X} \subset \mathbb{R}^d$ | Support of observations |
| $y \in \mathbb{R}$ | Target |
| $\mathcal{Y} \subset \mathbb{R}$ | Support of targets |

### Learned/ground-truth model

| | | |
|---|---|---|
| $\boldsymbol{W} \in \mathbb{R}^{k \times m}$ | | Ground-truth coefficients |
| $\hat{\boldsymbol{W}} \in \mathbb{R}^{k \times m}$ | | Learned coefficients |
| $\boldsymbol{\theta}$ | | Ground-truth parameters of the representation |
| $\hat{\boldsymbol{\theta}}$ | | Learned parameters of the representation |
| $\boldsymbol{f}_{\boldsymbol{\theta}} : \mathbb{R}^d \to \mathbb{R}^m$ | | Ground-truth representation |
| $\boldsymbol{f}_{\hat{\boldsymbol{\theta}}} : \mathbb{R}^d \to \mathbb{R}^m$ | | Learned representation |
| $\boldsymbol{\eta} \in \mathbb{R}^k$ | | Parameter of the distribution $p(y; \boldsymbol{\eta})$ |
| $\mathbb{P}_{\boldsymbol{W}}$ | | Distribution over ground-truth coefficient matrices $\boldsymbol{W}$ |
| $S$ | $:=$ | $\{j \in [m] \mid \boldsymbol{W}_{:j} \neq \boldsymbol{0}\}$ (support of $\boldsymbol{W}$) |
| $\mathbb{P}_{\boldsymbol{W}\mid S}$ | | Conditional distribution of $\boldsymbol{W}$ given $S$. |
| $p(S)$ | | Ground-truth distribution over possible supports $S$ |
| $\mathcal{S}$ | | Support of the distribution $p(S)$ |

### Optimization

| | | |
|---|---|---|
| $W$ | | Primal variable |
| $\Lambda$ | | Dual variable |
| $h^* : \boldsymbol{a}$ | $\mapsto$ | $\sup_{\boldsymbol{b} \in \mathbb{R}^d} \langle \boldsymbol{a}, \boldsymbol{b} \rangle - h(\boldsymbol{b})$, Fenchel conjugate of the function $h : \mathbb{R}^d \to \mathbb{R}$ |
| $f \square g : \boldsymbol{a}$ | $\mapsto$ | $\min_b f(\boldsymbol{a} - \boldsymbol{b}) + g(\boldsymbol{b})$, inf-convolution of the functions $f$ and $g$ |
| $\text{BST} : (\boldsymbol{a}, \tau)$ | $\mapsto$ | $(1 - \tau/\|\boldsymbol{a}\|)_+ \, \boldsymbol{a}$, block soft-thresholding operator |

## A    PROOFS OF SECTION 2.1

**Proposition 1** (MLE Invariance to Invertible Linear Transformations of the Features). *Let $\hat{\boldsymbol{W}}_n^{(\hat{\boldsymbol{\theta}})}$ and $\hat{\boldsymbol{W}}_n^{(\boldsymbol{\theta})}$ be the solutions to Problem (1) with the representations $\boldsymbol{f}_{\hat{\boldsymbol{\theta}}}$ and $\boldsymbol{f}_{\boldsymbol{\theta}}$, respectively (which we assume are unique). If there exists an invertible matrix $\boldsymbol{L}$ such that, $\forall \boldsymbol{x} \in \mathcal{X}$, $\boldsymbol{f}_{\hat{\boldsymbol{\theta}}}(\boldsymbol{x}) = \boldsymbol{L}\boldsymbol{f}_{\boldsymbol{\theta}}(\boldsymbol{x})$; then we have, $\forall \boldsymbol{x} \in \mathcal{X}$, $\hat{\boldsymbol{W}}_n^{(\hat{\boldsymbol{\theta}})}\boldsymbol{f}_{\hat{\boldsymbol{\theta}}}(\boldsymbol{x}) = \hat{\boldsymbol{W}}_n^{(\boldsymbol{\theta})}\boldsymbol{f}_{\boldsymbol{\theta}}(\boldsymbol{x})$.*

*Proof.* By definition of $\hat{\boldsymbol{W}}^{(\hat{\boldsymbol{\theta}})}$, we have that, for all $\hat{\boldsymbol{W}} \in \mathbb{R}^{k \times m}$,

$$\sum_{(\boldsymbol{x},y) \in \mathcal{D}} \log p(y; \hat{\boldsymbol{W}}^{(\hat{\boldsymbol{\theta}})} \boldsymbol{f}_{\hat{\boldsymbol{\theta}}}(\boldsymbol{x})) \geq \sum_{(\boldsymbol{x},y) \in \mathcal{D}} \log p(y; \hat{\boldsymbol{W}} \boldsymbol{f}_{\hat{\boldsymbol{\theta}}}(\boldsymbol{x})) \tag{7}$$

$$\sum_{(\boldsymbol{x},y) \in \mathcal{D}} \log p(y; \hat{\boldsymbol{W}}^{(\hat{\boldsymbol{\theta}})} \boldsymbol{L} \boldsymbol{f}_{\boldsymbol{\theta}}(\boldsymbol{x})) \geq \sum_{(\boldsymbol{x},y) \in \mathcal{D}} \log p(y; \hat{\boldsymbol{W}} \boldsymbol{L} \boldsymbol{f}_{\boldsymbol{\theta}}(\boldsymbol{x})). \tag{8}$$

Because $\mathbb{R}^{k \times m} \boldsymbol{L} = \mathbb{R}^{k \times m}$, we have that, for all $\hat{\boldsymbol{W}} \in \mathbb{R}^{k \times m}$,

$$\sum_{(\boldsymbol{x}, y) \in \mathcal{D}} \log p(y; \hat{\boldsymbol{W}}^{(\hat{\boldsymbol{\theta}})} \boldsymbol{L} \boldsymbol{f}_{\boldsymbol{\theta}}(\boldsymbol{x})) \geq \sum_{(\boldsymbol{x}, y) \in \mathcal{D}} \log p(y; \hat{\boldsymbol{W}} \boldsymbol{f}_{\boldsymbol{\theta}}(\boldsymbol{x})), \tag{9}$$

which is to say that $\hat{\boldsymbol{W}}^{(\boldsymbol{\theta})} = \hat{\boldsymbol{W}}^{(\hat{\boldsymbol{\theta}})} \boldsymbol{L}$, or put differently, $\hat{\boldsymbol{W}}^{(\hat{\boldsymbol{\theta}})} = \hat{\boldsymbol{W}}^{(\boldsymbol{\theta})} \boldsymbol{L}^{-1}$. It implies

$$\hat{\boldsymbol{W}}^{(\hat{\boldsymbol{\theta}})} \boldsymbol{f}_{\hat{\boldsymbol{\theta}}}(\boldsymbol{x}) = \hat{\boldsymbol{W}}^{(\boldsymbol{\theta})} \boldsymbol{L}^{-1} \boldsymbol{L} \boldsymbol{f}_{\boldsymbol{\theta}}(\boldsymbol{x}) = \hat{\boldsymbol{W}}^{(\boldsymbol{\theta})} \boldsymbol{f}_{\boldsymbol{\theta}}(\boldsymbol{x}), \tag{10}$$

which is what we wanted to show. $\qquad \square$

**Proposition 2** (Population MLE for Linearly Entangled Representations). *Let $\hat{\boldsymbol{W}}_{\infty}^{(\hat{\boldsymbol{\theta}})}$ be the solution of the population-based MLE, $\arg\max_{\tilde{\boldsymbol{W}}} \mathbb{E}_{p(\boldsymbol{x}, y)} \log p(y; \tilde{\boldsymbol{W}} \boldsymbol{f}_{\hat{\boldsymbol{\theta}}}(\boldsymbol{x}))$ (assumed to be unique). Suppose $\boldsymbol{f}_{\hat{\boldsymbol{\theta}}}$ is linearly equivalent to $\boldsymbol{f}_{\boldsymbol{\theta}}$, and Assumption 1 holds, then, $\hat{\boldsymbol{W}}_{\infty}^{(\hat{\boldsymbol{\theta}})} = \boldsymbol{W} \boldsymbol{L}^{-1}$.*

*Proof.* By definition of $\hat{\boldsymbol{W}}_{\infty}^{(\hat{\boldsymbol{\theta}})}$, we have that, for all $\tilde{\boldsymbol{W}} \in \mathbb{R}^{k \times m}$,

$$\mathbb{E}_{p(\boldsymbol{x}, y)} \log p(y; \hat{\boldsymbol{W}}_{\infty}^{(\hat{\boldsymbol{\theta}})} \boldsymbol{f}_{\hat{\boldsymbol{\theta}}}(\boldsymbol{x})) \geq \mathbb{E}_{p(\boldsymbol{x}, y)} \log p(y; \tilde{\boldsymbol{W}} \boldsymbol{f}_{\hat{\boldsymbol{\theta}}}(\boldsymbol{x})) \tag{11}$$

$$\mathbb{E}_{p(\boldsymbol{x}, y)} \log p(y; \hat{\boldsymbol{W}}_{\infty}^{(\hat{\boldsymbol{\theta}})} \boldsymbol{L} \boldsymbol{f}_{\boldsymbol{\theta}}(\boldsymbol{x})) \geq \mathbb{E}_{p(\boldsymbol{x}, y)} \log p(y; \tilde{\boldsymbol{W}} \boldsymbol{L} \boldsymbol{f}_{\boldsymbol{\theta}}(\boldsymbol{x})). \tag{12}$$

In particular, the inequality holds for $\tilde{\boldsymbol{W}} := \boldsymbol{W} \boldsymbol{L}^{-1}$, which yields

$$\mathbb{E}_{p(\boldsymbol{x}, y)} \log p(y; \hat{\boldsymbol{W}}_{\infty}^{(\hat{\boldsymbol{\theta}})} \boldsymbol{L} \boldsymbol{f}_{\boldsymbol{\theta}}(\boldsymbol{x})) \geq \mathbb{E}_{p(\boldsymbol{x}, y)} \log p(y; \boldsymbol{W} \boldsymbol{f}_{\boldsymbol{\theta}}(\boldsymbol{x})) \tag{13}$$

$$0 \geq \mathbb{E}_{p(\boldsymbol{x}, y)} \left[ \log p(y; \boldsymbol{W} \boldsymbol{f}_{\boldsymbol{\theta}}(\boldsymbol{x})) - \log p(y; \hat{\boldsymbol{W}}_{\infty}^{(\hat{\boldsymbol{\theta}})} \boldsymbol{L} \boldsymbol{f}_{\boldsymbol{\theta}}(\boldsymbol{x})) \right] \tag{14}$$

$$0 \geq \mathbb{E}_{p(\boldsymbol{x})} \mathrm{KL}(p(y; \boldsymbol{W} \boldsymbol{f}_{\boldsymbol{\theta}}(\boldsymbol{x})) \,||\, p(y; \hat{\boldsymbol{W}}_{\infty}^{(\hat{\boldsymbol{\theta}})} \boldsymbol{L} \boldsymbol{f}_{\boldsymbol{\theta}}(\boldsymbol{x}))). \tag{15}$$

Since the KL is always non-negative, we have that,

$$\mathbb{E}_{p(\boldsymbol{x})} \mathrm{KL}(p(y; \boldsymbol{W} \boldsymbol{f}_{\boldsymbol{\theta}}(\boldsymbol{x})) \,||\, p(y; \hat{\boldsymbol{W}}_{\infty}^{(\hat{\boldsymbol{\theta}})} \boldsymbol{L} \boldsymbol{f}_{\boldsymbol{\theta}}(\boldsymbol{x}))) = 0, \tag{16}$$

which in turn implies

$$\mathbb{E}_{p(\boldsymbol{x}, y)} \log p(y; \hat{\boldsymbol{W}}_{\infty}^{(\hat{\boldsymbol{\theta}})} \boldsymbol{L} \boldsymbol{f}_{\boldsymbol{\theta}}(\boldsymbol{x})) = \mathbb{E}_{p(\boldsymbol{x}, y)} \log p(y; \boldsymbol{W} \boldsymbol{f}_{\boldsymbol{\theta}}(\boldsymbol{x})) \tag{17}$$

$$\mathbb{E}_{p(\boldsymbol{x}, y)} \log p(y; \hat{\boldsymbol{W}}_{\infty}^{(\hat{\boldsymbol{\theta}})} \boldsymbol{L} \boldsymbol{f}_{\boldsymbol{\theta}}(\boldsymbol{x})) = \mathbb{E}_{p(\boldsymbol{x}, y)} \log p(y; \boldsymbol{W} \boldsymbol{L}^{-1} \boldsymbol{L} \boldsymbol{f}_{\boldsymbol{\theta}}(\boldsymbol{x})) \tag{18}$$

$$\mathbb{E}_{p(\boldsymbol{x}, y)} \log p(y; \hat{\boldsymbol{W}}_{\infty}^{(\hat{\boldsymbol{\theta}})} \boldsymbol{f}_{\hat{\boldsymbol{\theta}}}(\boldsymbol{x})) = \mathbb{E}_{p(\boldsymbol{x}, y)} \log p(y; \boldsymbol{W} \boldsymbol{L}^{-1} \boldsymbol{f}_{\hat{\boldsymbol{\theta}}}(\boldsymbol{x})) \tag{19}$$

$$\tag{20}$$

Since the solution to the population MLE from Proposition 2 is assumed to be unique, this equality holds if and only if $\hat{\boldsymbol{W}}_{\infty}^{(\hat{\boldsymbol{\theta}})} = \boldsymbol{W} \boldsymbol{L}^{-1}$. $\qquad \square$

## B    IDENTIFIABILITY THEORY

The following lemma will be important for proving Theorem 3. The argument is taken from Lachapelle et al. (2022).

**Lemma 1** (Sparsity pattern of an invertible matrix contains a permutation). *Let $\boldsymbol{L} \in \mathbb{R}^{m \times m}$ be an invertible matrix. Then, there exists a permutation $\sigma$ such that $\boldsymbol{L}_{i, \sigma(i)} \neq 0$ for all $i$.*

*Proof.* Since the matrix $\boldsymbol{L}$ is invertible, its determinant is non-zero, i.e.

$$\det(\boldsymbol{L}) := \sum_{\sigma \in \mathfrak{S}_m} \mathrm{sign}(\sigma) \prod_{i=1}^{m} \boldsymbol{L}_{i, \sigma(i)} \neq 0, \tag{21}$$

where $\mathfrak{S}_m$ is the set of $m$-permutations. This equation implies that at least one term of the sum is non-zero, meaning there exists $\sigma \in \mathfrak{S}_m$ such that for all $i \in [m]$, $\boldsymbol{L}_{i, \sigma(i)} \neq 0$. $\qquad \square$

For all $\boldsymbol{W} \in \mathcal{W}$, we are going to denote by $\hat{\boldsymbol{W}}^{(\boldsymbol{W})}$ some estimator of $\boldsymbol{W}$. The following result provides conditions under which if $\hat{\boldsymbol{W}}^{(\boldsymbol{W})}$ allows a perfect fit of the ground-truth distribution $p(y \mid \boldsymbol{x}, \boldsymbol{W})$, then the representation $\boldsymbol{f_\theta}$ and the parameter $\boldsymbol{W}$ are identified up to an invertible linear transformation. Many works have showed similar results in various context (Hyvärinen & Morioka, 2016; Khemakhem et al., 2020a; Roeder et al., 2021; Ahuja et al., 2022c). We reuse some of their proof techniques.

**Theorem 2** (Linear identifiability). *Let $\hat{\boldsymbol{W}}^{(\cdot)} : \mathcal{W} \to \mathbb{R}^{k \times m}$. Suppose Assumptions 2 to 5 hold and that, for all $\boldsymbol{W} \in \mathcal{W}$, $\boldsymbol{x} \in \mathcal{X}$ and $y \in \mathcal{Y}$, the following holds*

$$p(y; \hat{\boldsymbol{W}}^{(\boldsymbol{W})} \boldsymbol{f_{\hat{\theta}}}(\boldsymbol{x})) = p(y; \boldsymbol{W} \boldsymbol{f_\theta}(\boldsymbol{x})) \ . \tag{22}$$

*Then, there exists an invertible matrix $\boldsymbol{L} \in \mathbb{R}^{m \times m}$ such that, for all $\boldsymbol{x} \in \mathcal{X}$, $\boldsymbol{f_\theta}(\boldsymbol{x}) = \boldsymbol{L} \boldsymbol{f_{\hat{\theta}}}(\boldsymbol{x})$ and such that, for all $\boldsymbol{W} \in \mathcal{W}$, $\hat{\boldsymbol{W}}^{(\boldsymbol{W})} = \boldsymbol{W} \boldsymbol{L}$*

*Proof.* By Assumption 3, Equation (23) implies that $\boldsymbol{W} \boldsymbol{f_\theta}(\boldsymbol{x}) = \hat{\boldsymbol{W}}^{(\boldsymbol{W})} \boldsymbol{f_{\hat{\theta}}}(\boldsymbol{x})$. Assumption 5 ensures that we can construct an invertible matrix $\boldsymbol{U} := \begin{bmatrix} \boldsymbol{W}_{i_1,:}^{(1)} \\ \vdots \\ \boldsymbol{W}_{i_{d_z},:}^{(d_z)} \end{bmatrix}$. Construct analogously $\hat{\boldsymbol{U}} :=$

$\begin{bmatrix} \hat{\boldsymbol{W}}_{i_1,:}^{(\boldsymbol{W}^{(1)})} \\ \vdots \\ \hat{\boldsymbol{W}}_{i_{d_z},:}^{(\boldsymbol{W}^{(d_z)})} \end{bmatrix}$. This allows us to write $\boldsymbol{U} \boldsymbol{f_\theta}(\boldsymbol{x}) = \hat{\boldsymbol{U}} \boldsymbol{f_{\hat{\theta}}}(\boldsymbol{x})$. Left-multiplying by $\boldsymbol{U}^{-1}$ on both sides

yields $\boldsymbol{f_\theta}(\boldsymbol{x}) = \boldsymbol{L} \boldsymbol{f_{\hat{\theta}}}(\boldsymbol{x})$, where $\boldsymbol{L} := \boldsymbol{U}^{-1} \hat{\boldsymbol{U}}$. Using the invertible matrix $\boldsymbol{F}$ from Assumption 4, we can thus write $\boldsymbol{F} = \boldsymbol{L} \hat{\boldsymbol{F}}$ where we defined $\hat{\boldsymbol{F}} := [\boldsymbol{f_{\hat{\theta}}}(\boldsymbol{x}^{(1)}), \cdots, \boldsymbol{f_{\hat{\theta}}}(\boldsymbol{x}^{(d_z)})]$. Since $\boldsymbol{F}$ is invertible, so are $\boldsymbol{L}$ and $\hat{\boldsymbol{F}}$.

By substituting $\boldsymbol{F} = \boldsymbol{L} \hat{\boldsymbol{F}}$ in $\boldsymbol{W} \boldsymbol{F} = \hat{\boldsymbol{W}}^{(\boldsymbol{W})} \hat{\boldsymbol{F}}$, we obtain $\boldsymbol{W} \boldsymbol{L} \hat{\boldsymbol{F}} = \hat{\boldsymbol{W}}^{(\boldsymbol{W})} \hat{\boldsymbol{F}}$. By right-multiplying both sides by $\hat{\boldsymbol{F}}^{-1}$, we obtain $\boldsymbol{W} \boldsymbol{L} = \hat{\boldsymbol{W}}^{(\boldsymbol{W})}$. $\square$

The following theorem is where most of the theoretical contribution of this work lies. Note that Theorem 1, from the main text, is a straightforward application of this result.

**Theorem 3.** *(Disentanglement via task sparsity) Let $\hat{\boldsymbol{W}}^{(\cdot)} : \mathcal{W} \to \mathbb{R}^{k \times m}$. Suppose Assumptions 3 to 7 hold and that, for all $\boldsymbol{W} \in \mathcal{W}$, $\boldsymbol{x} \in \mathcal{X}$ and $y \in \mathcal{Y}$, the following holds*

$$p(y; \hat{\boldsymbol{W}}^{(\boldsymbol{W})} \boldsymbol{f_{\hat{\theta}}}(\boldsymbol{x})) = p(y; \boldsymbol{W} \boldsymbol{f_\theta}(\boldsymbol{x})) \ . \tag{23}$$

*Morevover, assume that $\mathbb{E} \|\hat{\boldsymbol{W}}^{(\boldsymbol{W})}\|_{2,0} \leq \mathbb{E} \|\boldsymbol{W}\|_{2,0}$, where both expectations are taken w.r.t. $\mathbb{P}_{\boldsymbol{W}}$ and $\|\boldsymbol{W}\|_{2,0} := \sum_{j=1}^m \mathbb{1}(\boldsymbol{W}_{:j} \neq \boldsymbol{0})$ with $\mathbb{1}(\cdot)$ the indicator function. Then, $\boldsymbol{f_{\hat{\theta}}}$ is disentangled w.r.t. $\boldsymbol{f_\theta}$ (Definition 1).*

*Proof.* First of all, by Assumptions 3 to 5, we can apply Theorem 2 to conclude that $\boldsymbol{f_\theta}(\boldsymbol{x}) = \boldsymbol{L} \boldsymbol{f_{\hat{\theta}}}(\boldsymbol{x})$ and $\boldsymbol{W} \boldsymbol{L} = \hat{\boldsymbol{W}}^{(\boldsymbol{W})}$ for some invertible matrix $\boldsymbol{L}$.

We can thus write $\mathbb{E} \|\boldsymbol{W} \boldsymbol{L}\|_{2,0} \leq \mathbb{E} \|\boldsymbol{W}\|_{2,0}$.

We can write

$$\mathbb{E}\|\boldsymbol{W}\|_{2,0} = \mathbb{E}_{p(S)}\mathbb{E}[\sum_{j=1}^{m} \mathbb{1}(\boldsymbol{W}_{:j} \neq \boldsymbol{0}) \mid S] \tag{24}$$

$$= \mathbb{E}_{p(S)} \sum_{j=1}^{m} \mathbb{E}[\mathbb{1}(\boldsymbol{W}_{:j} \neq \boldsymbol{0}) \mid S] \tag{25}$$

$$= \mathbb{E}_{p(S)} \sum_{j=1}^{m} \mathbb{P}_{\boldsymbol{W}|S}[\boldsymbol{W}_{:j} \neq \boldsymbol{0}] \tag{26}$$

$$= \mathbb{E}_{p(S)} \sum_{j=1}^{m} \mathbb{1}(j \in S), \tag{27}$$

where the last step follows from the definition of $S$.

We now perform similar steps for $\mathbb{E}\|\boldsymbol{W}\boldsymbol{L}\|_{2,0}$:

$$\mathbb{E}\|\boldsymbol{W}\boldsymbol{L}\|_{2,0} = \mathbb{E}_{p(S)}\mathbb{E}[\sum_{j=1}^{m} \mathbb{1}(\boldsymbol{W}\boldsymbol{L}_{:j} \neq \boldsymbol{0}) \mid S] \tag{28}$$

$$= \mathbb{E}_{p(S)} \sum_{j=1}^{m} \mathbb{E}[\mathbb{1}(\boldsymbol{W}\boldsymbol{L}_{:j} \neq \boldsymbol{0}) \mid S] \tag{29}$$

$$= \mathbb{E}_{p(S)} \sum_{j=1}^{m} \mathbb{P}_{\boldsymbol{W}|S}[\boldsymbol{W}\boldsymbol{L}_{:j} \neq \boldsymbol{0}] \tag{30}$$

$$= \mathbb{E}_{p(S)} \sum_{j=1}^{m} \mathbb{P}_{\boldsymbol{W}|S}[\boldsymbol{W}_{:S}\boldsymbol{L}_{S,j} \neq \boldsymbol{0}]. \tag{31}$$

Notice that

$$\mathbb{P}_{\boldsymbol{W}|S}[\boldsymbol{W}_{:S}\boldsymbol{L}_{S,j} \neq \boldsymbol{0}] = 1 - \mathbb{P}_{\boldsymbol{W}|S}[\boldsymbol{W}_{:S}\boldsymbol{L}_{S,j} = \boldsymbol{0}] \tag{32}$$

Let $N_j$ be the support of $\boldsymbol{L}_{:j}$, i.e. $N_j := \{i \in [m] \mid \boldsymbol{L}_{i,j} \neq 0\}$. When $S \cap N_j = \emptyset$, $\boldsymbol{L}_{S,j} = \boldsymbol{0}$ and thus $\mathbb{P}_{\boldsymbol{W}|S}[\boldsymbol{W}_{:S}\boldsymbol{L}_{S,j} = \boldsymbol{0}] = 1$. When $S \cap N_j \neq \emptyset$, $\boldsymbol{L}_{S,j} \neq \boldsymbol{0}$, by Assumption 6 we have that $\mathbb{P}_{\boldsymbol{W}|S}[\boldsymbol{W}_{:S}\boldsymbol{L}_{S,j} = \boldsymbol{0}] = 0$. Thus

$$\mathbb{P}_{\boldsymbol{W}|S}[\boldsymbol{W}_{:S}\boldsymbol{L}_{S,j} \neq \boldsymbol{0}] = 1 - \mathbb{1}(S \cap N_j = \emptyset) \tag{33}$$

$$= \mathbb{1}(S \cap N_j \neq \emptyset), \tag{34}$$

which allows us to write

$$\mathbb{E}\|\boldsymbol{W}\boldsymbol{L}\|_{2,0} = \mathbb{E}_{p(S)} \sum_{j=1}^{m} \mathbb{1}(S \cap N_j \neq \emptyset). \tag{35}$$

We thus have that

$$\mathbb{E}\|\boldsymbol{W}\boldsymbol{L}\|_{2,0} \leq \mathbb{E}\|\boldsymbol{W}\|_{2,0} \tag{36}$$

$$\mathbb{E}_{p(S)} \sum_{j=1}^{m} \mathbb{1}(S \cap N_j \neq \emptyset) \leq \mathbb{E}_{p(S)} \sum_{j=1}^{m} \mathbb{1}(j \in S). \tag{37}$$

Since $\boldsymbol{L}$ is invertible, by Lemma 1, there exists a permutation $\sigma : [m] \to [m]$ such that, for all $j \in [m]$, $\boldsymbol{L}_{j,\sigma(j)} \neq 0$. In other words, for all $j \in [m]$, $j \in N_{\sigma(j)}$. Of course we can permute the terms of the l.h.s. of eq. (37), which yields

$$\mathbb{E}_{p(S)} \sum_{j=1}^{m} \mathbb{1}(S \cap N_{\sigma(j)} \neq \emptyset) \leq \mathbb{E}_{p(S)} \sum_{j=1}^{m} \mathbb{1}(j \in S) \tag{38}$$

$$\mathbb{E}_{p(S)} \sum_{j=1}^{m} \left( \mathbb{1}(S \cap N_{\sigma(j)} \neq \emptyset) - \mathbb{1}(j \in S) \right) \leq 0. \tag{39}$$

We notice that each term $\mathbb{1}(S \cap N_{\sigma(j)} \neq \emptyset) - \mathbb{1}(j \in S) \geq 0$ since whenever $j \in S$, we also have that $j \in S \cap N_{\sigma(j)}$ (recall $j \in N_{\sigma(j)}$). Thus, the l.h.s. of eq. (39) is a sum of non-negative terms which is itself non-positive. This means that every term in the sum is zero:

$$\forall S \in \mathcal{S}, \ \forall j \in [m], \ \mathbb{1}(S \cap N_{\sigma(j)} \neq \emptyset) = \mathbb{1}(j \in S). \tag{40}$$

Importantly,

$$\forall j \in [m], \ \forall S \in \mathcal{S}, \ j \notin S \implies S \cap N_{\sigma(j)} = \emptyset, \tag{41}$$

and since $S \cap N_{\sigma(j)} = \emptyset \iff N_{\sigma(j)} \subseteq S^c$ we have that

$$\forall j \in [m], \ \forall S \in \mathcal{S}, \ j \notin S \implies N_{\sigma(j)} \subseteq S^c \tag{42}$$

$$\forall j \in [m], \ N_{\sigma(j)} \subseteq \bigcap_{S \in \mathcal{S} | j \notin S} S^c. \tag{43}$$

By Assumption 7, we have that $\bigcup_{S \in \mathcal{S} | j \notin S} S = [m] \setminus \{j\}$. By taking the complement on both sides and using De Morgan's law, we get $\bigcap_{S \in \mathcal{S} | j \notin S} S^c = \{j\}$, which implies that $N_{\sigma(j)} = \{j\}$ by Equation (43). Thus, $\boldsymbol{L} = \boldsymbol{D}\boldsymbol{P}$ where $\boldsymbol{D}$ is an invertible diagonal matrix and $\boldsymbol{P}$ is a permutation matrix. $\square$

## B.1 PROOF OF THEOREM 1

Before presenting Theorem 1 from the main text, we first present a variation of it where we constrain $\mathbb{E}\|\hat{\boldsymbol{W}}^{(\boldsymbol{W})}\|_{2,0}$ to be smaller than $\mathbb{E}\|\boldsymbol{W}\|_{2,0}$. We note that this is weaker than imposing $\|\hat{\boldsymbol{W}}^{(\boldsymbol{W})}\|_{2,0} \leq \|\boldsymbol{W}\|_{2,0}$ for all $\boldsymbol{W} \in \mathcal{W}$, as is the case in Problem (3) of Theorem 1.

**Theorem 4** (Sparse multitask learning for disentanglement). *Let $\hat{\boldsymbol{\theta}}$ be a minimizer of*

$$\min_{\hat{\boldsymbol{\theta}}} \ \mathbb{E}_{\mathbb{P}_{\boldsymbol{W}}} \mathbb{E}_{p(\boldsymbol{x},y|\boldsymbol{W})} - \log p(y; \hat{\boldsymbol{W}}^{(\boldsymbol{W})} \boldsymbol{f}_{\hat{\boldsymbol{\theta}}}(\boldsymbol{x}))$$

$$\text{s.t.} \quad \forall \boldsymbol{W} \in \mathcal{W}, \hat{\boldsymbol{W}}^{(\boldsymbol{W})} \in \arg\min_{\tilde{\boldsymbol{W}}} \mathbb{E}_{p(\boldsymbol{x},y|\boldsymbol{W})} - \log p(y; \tilde{\boldsymbol{W}} \boldsymbol{f}_{\hat{\boldsymbol{\theta}}}(\boldsymbol{x})) \tag{44}$$

$$\mathbb{E}_{\mathbb{P}_{\boldsymbol{W}}} \|\hat{\boldsymbol{W}}^{(\boldsymbol{W})}\|_{2,0} \leq \mathbb{E}_{\mathbb{P}_{\boldsymbol{W}}} \|\boldsymbol{W}\|_{2,0} \ .$$

*Then, under Assumptions 2 to 7, $\boldsymbol{f}_{\hat{\boldsymbol{\theta}}}$ is disentangled w.r.t. $\boldsymbol{f}_{\boldsymbol{\theta}}$ (Definition 1).*

*Proof.* First, notice that

$$0 \leq \mathbb{E}_{\mathbb{P}_{\boldsymbol{W}}} \mathbb{E}_{p(\boldsymbol{x}|\boldsymbol{W})} \text{KL}(p(y; \boldsymbol{W} \boldsymbol{f}_{\boldsymbol{\theta}}(\boldsymbol{x})) \ || \ p(y; \hat{\boldsymbol{W}}^{(\boldsymbol{W})} \boldsymbol{f}_{\hat{\boldsymbol{\theta}}}(\boldsymbol{x}))) \tag{45}$$

$$\mathbb{E}_{\mathbb{P}_{\boldsymbol{W}}} \mathbb{E}_{p(\boldsymbol{x},y|\boldsymbol{W})} - \log p(y; \boldsymbol{W} \boldsymbol{f}_{\boldsymbol{\theta}}(\boldsymbol{x})) \leq \mathbb{E}_{\mathbb{P}_{\boldsymbol{W}}} \mathbb{E}_{p(\boldsymbol{x},y|\boldsymbol{W})} - \log p(y; \hat{\boldsymbol{W}}^{(\boldsymbol{W})} \boldsymbol{f}_{\hat{\boldsymbol{\theta}}}(\boldsymbol{x})). \tag{46}$$

For a fixed value of $\boldsymbol{x}$ and $\boldsymbol{W}$, it is well known that $\text{KL}(p(y; \boldsymbol{W} \boldsymbol{f}_{\boldsymbol{\theta}}(\boldsymbol{x})) \ || \ p(y; \hat{\boldsymbol{W}}^{(\boldsymbol{W})} \boldsymbol{f}_{\hat{\boldsymbol{\theta}}}(\boldsymbol{x}))) = 0$ if and only if, for all $y \in \mathcal{Y}$, $p(y; \boldsymbol{W} \boldsymbol{f}_{\boldsymbol{\theta}}(\boldsymbol{x})) = p(y; \hat{\boldsymbol{W}}^{(\boldsymbol{W})} \boldsymbol{f}_{\hat{\boldsymbol{\theta}}}(\boldsymbol{x}))$. By Assumption 3, this is equivalent to $\boldsymbol{W} \boldsymbol{f}_{\boldsymbol{\theta}}(\boldsymbol{x}) = \hat{\boldsymbol{W}}^{(\boldsymbol{W})} \boldsymbol{f}_{\hat{\boldsymbol{\theta}}}(\boldsymbol{x})$. Thus, for the equality to hold in eq. (45), we need $\boldsymbol{W} \boldsymbol{f}_{\boldsymbol{\theta}}(\boldsymbol{x}) = \hat{\boldsymbol{W}}^{(\boldsymbol{W})} \boldsymbol{f}_{\hat{\boldsymbol{\theta}}}(\boldsymbol{x})$ everywhere. Of course, the global minimum can be achieved by respecting $\mathbb{E}_{\mathbb{P}_{\boldsymbol{W}}} \|\hat{\boldsymbol{W}}^{(\boldsymbol{W})}\|_{2,0} \leq \mathbb{E}_{\mathbb{P}_{\boldsymbol{W}}} \|\boldsymbol{W}\|_{2,0}$, simply by setting $\hat{\boldsymbol{\theta}} := \boldsymbol{\theta}$ and $\hat{\boldsymbol{W}}^{(\boldsymbol{W})} := \boldsymbol{W}$.

The above implies that if $\hat{\boldsymbol{\theta}}$ is some minimizer of Problem (44), we must have that $\boldsymbol{W} \boldsymbol{f}_{\boldsymbol{\theta}}(\boldsymbol{x}) = \hat{\boldsymbol{W}}^{(\boldsymbol{W})} \boldsymbol{f}_{\hat{\boldsymbol{\theta}}}(\boldsymbol{x})$ everywhere and $\mathbb{E}_{\mathbb{P}_{\boldsymbol{W}}} \|\hat{\boldsymbol{W}}^{(\boldsymbol{W})}\|_0 \leq \mathbb{E}_{\mathbb{P}_{\boldsymbol{W}}} \|\boldsymbol{W}\|_0$. Thus, Theorem 3 implies the desired conclusion. $\square$

Based on Theorem 4, we can slightly adjust the argument to prove Theorem 1 from the main text.

**Theorem 1** (Sparse multi-task learning for disentanglement). *Let $\hat{\boldsymbol{\theta}}$ be a minimizer of*

$$\min_{\hat{\boldsymbol{\theta}}} \ \mathbb{E}_{\mathbb{P}_{\boldsymbol{W}}} \mathbb{E}_{p(\boldsymbol{x},y|\boldsymbol{W})} - \log p(y; \hat{\boldsymbol{W}}^{(\boldsymbol{W})} \boldsymbol{f}_{\hat{\boldsymbol{\theta}}}(\boldsymbol{x}))$$

$$\text{s.t.} \quad \forall \boldsymbol{W} \in \mathcal{W}, \hat{\boldsymbol{W}}^{(\boldsymbol{W})} \in \arg\min_{\substack{\tilde{\boldsymbol{W}} \text{ s.t.} \\ \|\tilde{\boldsymbol{W}}\|_{2,0} \leq \|\boldsymbol{W}\|_{2,0}}} \mathbb{E}_{p(\boldsymbol{x},y|\boldsymbol{W})} - \log p(y; \tilde{\boldsymbol{W}} \boldsymbol{f}_{\hat{\boldsymbol{\theta}}}(\boldsymbol{x})) \ . \tag{3}$$

*Then, under Assumptions 2 to 7, $\boldsymbol{f}_{\hat{\boldsymbol{\theta}}}$ is disentangled w.r.t. $\boldsymbol{f}_{\boldsymbol{\theta}}$ (Definition 1).*

*Proof.* The first part of the argument in the proof of Theorem 4 applies here as well, meaning: for the equality to hold in eq. (45), we need $\boldsymbol{W}\boldsymbol{f}_{\boldsymbol{\theta}}(\boldsymbol{x}) = \hat{\boldsymbol{W}}^{(\boldsymbol{W})}\boldsymbol{f}_{\hat{\boldsymbol{\theta}}}(\boldsymbol{x})$ everywhere. This global minimum can be achieved by respecting $\|\hat{\boldsymbol{W}}^{(\boldsymbol{W})}\|_{2,0} \le \|\boldsymbol{W}\|_{2,0}$ for all $\boldsymbol{W} \in \mathcal{W}$ simply by setting $\hat{\boldsymbol{\theta}} := \boldsymbol{\theta}$ and $\hat{\boldsymbol{W}}^{(\boldsymbol{W})} := \boldsymbol{W}$.

This means that if $\hat{\boldsymbol{\theta}}$ is some minimizer of Problem (3), we must have that $\boldsymbol{W}\boldsymbol{f}_{\boldsymbol{\theta}}(\boldsymbol{x}) = \hat{\boldsymbol{W}}^{(\boldsymbol{W})}\boldsymbol{f}_{\hat{\boldsymbol{\theta}}}(\boldsymbol{x})$ holds everywhere and that, for all $\boldsymbol{W} \in \mathcal{W}$, $\|\hat{\boldsymbol{W}}^{(\boldsymbol{W})}\|_{2,0} \le \|\boldsymbol{W}\|_{2,0}$. Of course, this means $\mathbb{E}_{\mathbb{P}_{\boldsymbol{W}}}\|\hat{\boldsymbol{W}}^{(\boldsymbol{W})}\|_0 \le \mathbb{E}_{\mathbb{P}_{\boldsymbol{W}}}\|\boldsymbol{W}\|_0$, which allows us to apply Theorem 3 to obtain the desired conclusion. $\square$

### B.2 A DISTRIBUTION WITHOUT DENSITY SATISFYING ASSUMPTION 6

Interestingly, there are distributions over $\boldsymbol{W}_{1,S} \mid S$ that do not have a density w.r.t. the Lebesgue measure, but still satisfy Assumption 6. This is the case, e.g., when $\boldsymbol{W}_{1,S} \mid S$ puts uniform mass over a $(|S| - 1)$-dimensional sphere embedded in $\mathbb{R}^{|S|}$ and centered at zero. In that case, for all $\boldsymbol{a} \in \mathbb{R}^{|S|}\backslash\{0\}$, the intersection of $\text{span}\{\boldsymbol{a}\}^{\perp}$, which is $(|S| - 1)$-dimensional, with the $(|S| - 1)$-dimensional sphere is $(|S| - 2)$-dimensional and thus has probability zero of occurring. One can certainly construct more exotic examples of measures satisfying Assumption 6 that concentrate mass on lower dimensional manifold.

## C  OPTIMIZATION DETAILS

### C.1  GROUP LASSO SVM DUAL

**Notation.** The Fenchel conjugate of a function $h : \mathbb{R}^d \to \mathbb{R}$ is written $h^*$ and is defined for any $y \in \mathbb{R}^d$, by $h^*(y) = \sup_{x \in \mathbb{R}^d}\langle x, y\rangle - h(x)$.

**Definition 2.** (Primal Group Lasso Soft-Margin Multiclass SVM.) *The primal problem of the group Lasso soft-margin multiclass SVM is defined as*

$$\min_{\boldsymbol{W} \in \mathbb{R}^{k \times m}} \mathcal{L}_{\text{in}}(\boldsymbol{W}; \boldsymbol{F}, \boldsymbol{Y}) := \sum_{i=1}^{n} \max_{l \in [k]} \left(1 + (\boldsymbol{W}_{\boldsymbol{y}_i:} - \boldsymbol{W}_{l:})\boldsymbol{F}_{i:} - \boldsymbol{Y}_{il}\right) + \lambda_1\|\boldsymbol{W}\|_{2,1} + \tfrac{\lambda_2}{2}\|\boldsymbol{W}\|^2 \tag{47}$$

**Proposition 3.** (Dual Group Lasso Soft-Margin Multiclass SVM.) *The dual of the inner problem with $\mathcal{L}_{\text{in}}$ as defined in (5) writes*

$$\min_{\boldsymbol{\Lambda} \in \mathbb{R}^{n \times k}} \frac{1}{\lambda_2} \sum_{j=1}^{m} \|\text{BST}\left((\boldsymbol{Y} - \boldsymbol{\Lambda})^{\top}\boldsymbol{F}_{:j}, \lambda_1\right)\|^2 + \langle \boldsymbol{Y}, \boldsymbol{\Lambda}\rangle + \sum_{i=1}^{n} \mathbb{1}_{\sum_{l=1}^{k} \boldsymbol{\Lambda}_{il} = 1} + \sum_{i=1}^{n}\sum_{l=1}^{k} \mathbb{1}_{\boldsymbol{\Lambda}_{il} \ge 0}, \tag{6}$$

*with* BST *the block soft-thresholding operator:* $\text{BST} : (\boldsymbol{a}, \tau) \mapsto (1 - \tau/\|\boldsymbol{a}\|)_{+}\boldsymbol{a}$, $\boldsymbol{F} \in \mathbb{R}^{n \times m}$ *the concatenation of $\{\boldsymbol{f}_{\hat{\boldsymbol{\theta}}}(x)\}_{(x,y) \in \mathcal{D}^{\text{train}}}$. In addition, the primal-dual link writes, for all $j \in [m]$, $\boldsymbol{W}_{:j} = \text{BST}\left((\boldsymbol{Y} - \boldsymbol{\Lambda})^{\top}\boldsymbol{F}_{:j}, \lambda_1\right)/\lambda_2$.*

The primal objective 47 can be hard to minimize with modern solvers. Moreover in few-shot learning applications, the number of features $m$ is usually much larger than the number of samples $n$ (in Lee et al. 2019, $m = 1.6 \cdot 10^4$ and $n \le 25$), hence we solve the dual of Problem (47).

*Proof of Proposition 3.* Let $g : \boldsymbol{u} \mapsto \lambda_1\|\boldsymbol{u}\| + \frac{\lambda_2}{2}\|\boldsymbol{u}\|^2$. Proof of Proposition 3 is composed of the following lemmas.

**Lemma 2.**     i) *The dual of Problem (47) is*

$$\min_{\boldsymbol{\Lambda} \in \mathbb{R}^{n \times k}} \sum_{j=1}^{m} g^*((\boldsymbol{Y} - \boldsymbol{\Lambda})^{\top}\boldsymbol{F}_{:j}) + \langle \boldsymbol{Y}, \boldsymbol{\Lambda}\rangle \tag{48}$$

$$\text{s.t.} \quad \forall i \in [n], \sum_{l=1}^{k} \boldsymbol{\Lambda}_{il} = 1, \quad \forall i \in [n], l \in [k], \boldsymbol{\Lambda}_{il} \ge 0,$$

*where $g^*$ is the Fenchel conjugate of the function $g$.*

ii) *The Fenchel conjugate of the function g writes*

$$\forall \boldsymbol{v} \in \mathbb{R}^K, \, g^*(\boldsymbol{v}) = \frac{1}{\lambda_2} \|\mathrm{BST}(\boldsymbol{v}, \lambda_1)\|^2 \; . \tag{49}$$

Lemmas 4 *i)* and 4 *ii)* yields Proposition 3.

*Proof of Lemma 4 i).* The Lagrangian of Problem (47) writes:

$$\mathcal{L}(\boldsymbol{W}, \boldsymbol{\xi}, \boldsymbol{\Lambda}) = \sum_{j=1}^{m} g(\boldsymbol{W}_{:j}) + \sum_i \boldsymbol{\xi}_i + \sum_{i=1}^{n} \sum_{l=1}^{k} (1 - \boldsymbol{\xi}_i - \boldsymbol{W}_{\boldsymbol{y}_i:} \cdot \boldsymbol{F}_{i:} + \boldsymbol{W}_{l:} \cdot \boldsymbol{F}_{i:} - \boldsymbol{Y}_{il}) \boldsymbol{\Lambda}_{il} \; . \tag{50}$$

$\partial_{\boldsymbol{\xi}} \mathcal{L}(\boldsymbol{W}, \boldsymbol{\xi}, \boldsymbol{\Lambda}) = 0$ yields $\forall i \in [n], \sum_{l=1}^{k} \boldsymbol{\Lambda}_{il} = 1$. Then the Lagrangian rewrites

$$\min_{\boldsymbol{W}} \min_{\boldsymbol{\xi}} \mathcal{L}(\boldsymbol{W}, \boldsymbol{\xi}, \boldsymbol{\Lambda}) = \min_{\boldsymbol{W}, \boldsymbol{\xi}} \sum_{j=1}^{m} g(\boldsymbol{W}_{:j}) + \sum_{i=1}^{n} \boldsymbol{\xi}_i + \sum_{i=1}^{n} \sum_{l=1}^{k} (-\boldsymbol{\xi}_i - \boldsymbol{W}_{\boldsymbol{y}_i:} \cdot \boldsymbol{F}_{i:} + \boldsymbol{W}_{l:} \cdot \boldsymbol{F}_{i:} - \boldsymbol{Y}_{il}) \boldsymbol{\Lambda}_{il}$$

$$= \sum_{j=1}^{m} \min_{\boldsymbol{W}_{:j}} g(\boldsymbol{W}_{:j}) - \underbrace{\sum_{i=1}^{n} \sum_{l=1}^{k} (\boldsymbol{F}_{i:} \boldsymbol{Y}_{il} - \boldsymbol{F}_{i:} \boldsymbol{\Lambda}_{il}) \boldsymbol{W}_{l:}}_{\substack{=\langle (\boldsymbol{Y} - \boldsymbol{\Lambda})^\top \boldsymbol{F}_{:j}, \boldsymbol{W}_{:j} \rangle \\ = -g^*((\boldsymbol{Y} - \boldsymbol{\Lambda})^\top \boldsymbol{F}_{:j})}} - \sum_{i=1}^{n} \sum_{l=1}^{k} \boldsymbol{Y}_{il} \boldsymbol{\Lambda}_{il} \; .$$

Then the dual problem writes:

$$\min_{\boldsymbol{\Lambda} \in \mathbb{R}^{n \times k}} \sum_{j=1}^{m} g^* \left( (\boldsymbol{Y} - \boldsymbol{\Lambda})^\top \boldsymbol{F}_{:j} \right) + \langle \boldsymbol{Y}, \boldsymbol{\Lambda} \rangle \tag{51}$$

$$\text{s. t.} \quad \forall i \in [n] \quad \sum_{l=1}^{k} \boldsymbol{\Lambda}_{il} = 1 \; , \forall i \in [n], l \in [k], \quad \boldsymbol{\Lambda}_{il} \geq 0 \; . \tag{52}$$

$$\square$$

*Proof of Lemma 4 ii).* Let $h : \boldsymbol{u} \mapsto \|\boldsymbol{u}\|_2 + \frac{\kappa}{2}\|\boldsymbol{u}\|^2$. The proof of Lemma 4 *i)* is done using the following steps.

**Lemma 3.**     i) $h^*(\boldsymbol{v}) = \frac{1}{2\kappa}\|\boldsymbol{v}\|_2^2 - \left(\frac{\kappa}{2}\|\cdot\|_2^2 \square \|\cdot\|_2\right)(\boldsymbol{v}/\kappa)$.

ii) $\left(\frac{\kappa}{2}\|\cdot\|_2^2 \square \|\cdot\|_2\right)(\boldsymbol{v}) = \frac{\kappa}{2}\|\boldsymbol{v}\|_2^2 - \frac{1}{2\kappa}\|\mathrm{BST}(\kappa\boldsymbol{v}, 1)\|^2$.

*Proof of Lemma 4 i).* With $\kappa = \lambda_2/\lambda_1$, the Fenchel transform of $h : \boldsymbol{w} \mapsto \|\boldsymbol{w}\|_2 + \kappa\|\boldsymbol{w}\|^2$.

$$h(\boldsymbol{u}) = \|\boldsymbol{u}\|_2 + \frac{\kappa}{2}\|\boldsymbol{u}\|_2^2$$

$$h^*(\boldsymbol{v}) = \sup_{\boldsymbol{w}} \left(\boldsymbol{v}^\top \boldsymbol{w} - \|\boldsymbol{w}\|_2 - \frac{\kappa}{2}\|\boldsymbol{w}\|_2^2\right)$$

$$= \frac{1}{2\kappa}\|\boldsymbol{v}\|_2^2 + \sup_{\boldsymbol{w}} \left(-\frac{\kappa}{2}\|\boldsymbol{w} - \boldsymbol{v}/\kappa\|_2^2 - \|\boldsymbol{w}\|_2\right)$$

$$= \frac{1}{2\kappa}\|\boldsymbol{v}\|_2^2 - \inf_{\boldsymbol{w}} \left(\frac{\kappa}{2}\|\boldsymbol{w} - \boldsymbol{v}/\kappa\|_2^2 + \|\boldsymbol{w}\|_2\right)$$

$$= \frac{1}{2\kappa}\|\boldsymbol{v}\|_2^2 - \left(\frac{\kappa}{2}\|\cdot\|_2^2 \square \|\cdot\|_2\right)(\boldsymbol{v}/\kappa) \; .$$

$$\square$$

*Proof of Lemma 4 ii).*

$$
\begin{aligned}
(\tfrac{\kappa}{2}\|\cdot\|_2^2 \square \|\cdot\|_2)(\boldsymbol{v}) &= (\tfrac{\kappa}{2}\|\cdot\|_2^2 \square \|\cdot\|_2)^{**}(\boldsymbol{v}) \\
&= (\tfrac{1}{2\kappa}\|\cdot\|_2^2 + \iota_{\mathcal{B}_2})^*(\boldsymbol{v}) \\
&= \sup_{\|\boldsymbol{w}\|_2 \le 1} \left(\boldsymbol{v}^\top \boldsymbol{w} - \tfrac{1}{2\kappa}\|\boldsymbol{w}\|_2^2\right) \\
&= \tfrac{\kappa}{2}\|\boldsymbol{v}\|^2 + \sup_{\|\boldsymbol{w}\|_2 \le 1} -\tfrac{1}{2\kappa}\|\kappa\boldsymbol{v} - \boldsymbol{w}\|_2^2 \\
&= \tfrac{\kappa}{2}\|\boldsymbol{v}\|^2 - \tfrac{1}{2\kappa}\|\mathrm{BST}(\kappa\boldsymbol{v}, 1)\|_2^2 \ .
\end{aligned}
$$

$\square$

$$
\begin{aligned}
g^*(\boldsymbol{u}) &= \lambda_1 h^*(\boldsymbol{u}/\lambda_1) \\
&= \frac{\lambda_1}{2\kappa}\|\mathrm{BST}(\boldsymbol{u}/\lambda_1, 1)\|^2 \\
&= \frac{\lambda_1^2}{2\lambda_2}\|\mathrm{BST}(\boldsymbol{u}/\lambda_1, 1)\|^2 \\
&= \frac{1}{\lambda_2}\|\mathrm{BST}(\boldsymbol{u}, \lambda_1)\|^2 \ .
\end{aligned}
$$

$\square$

$\square$

## D   EXPERIMENTAL DETAILS

### D.1   DISENTANGLED REPRESENTATION COUPLED WITH SPARSITY REGULARIZATION IMPROVES GENERALIZATION

We consider the following data generating process: We sample the ground-truth features $\boldsymbol{f_\theta}(\boldsymbol{x})$ from a Gaussian distribution $\mathcal{N}(\boldsymbol{0}, \boldsymbol{\Sigma})$ where $\boldsymbol{\Sigma} \in \mathbb{R}^{m \times m}$ and $\boldsymbol{\Sigma}_{i,j} = 0.9^{|i-j|}$. Moreover, the labels are given by $y = \boldsymbol{w} \cdot \boldsymbol{f_\theta}(\boldsymbol{x}) + \epsilon$ where $\boldsymbol{w} \in \mathbb{R}^m$, $\epsilon \sim \mathcal{N}(0, 0.04)$ and $m = 100$. The ground-truth weight vector $\boldsymbol{w}$ is sampled once from $\mathcal{N}(0, I_{m \times m})$ and mask some of its components to zero: we vary the fraction of meaningful features ($\ell/m$) from very sparse ($\ell/m = 5\%$) to less sparse ($\ell/m = 80\%$) settings. For each case, we study the sample complexity by varying the number of training samples from 25 to 150, but evaluating the generalization performance on a larger test dataset (1000 samples). To generate the entangled representations, we multiply the true latent variables $\boldsymbol{f_\theta}(\boldsymbol{x})$ by a randomly sampled orthogonal matrix $\boldsymbol{L}$, i.e., $\boldsymbol{f_{\hat{\theta}}}(\boldsymbol{x}) := \boldsymbol{L}\boldsymbol{f_\theta}(\boldsymbol{x})$. For the disentangled representation, we simply consider the true latents, i.e. $\boldsymbol{f_{\hat{\theta}}}(\boldsymbol{x}) := \boldsymbol{f_\theta}(\boldsymbol{x})$. Note that in principle we could have considered an invertible matrix $\boldsymbol{L}$ that is not orthogonal for the linearly entangled representation and a component-wise rescaling for the disentangled representation. The advantage of not doing so and opting for our approach is that the conditioning number of the covariance matrix of $\boldsymbol{f_{\hat{\theta}}}(\boldsymbol{x})$ is the same for both the entangled and the disentangled, hence offering a fairer comparison.

For both the case of entangled and disentangled representation, we solve the regression problem with Lasso and Ridge regression, where the associated hyperparameters (regularization strength) were inferred using 5-fold cross validation on the input training dataset. Using both lasso and ridge regression would help us to show the effect of encouraging sparsity.

In Figure 1 for the sparsest case ($\ell/m = 5\%$), we observe that that Disentangled-Lasso approach has the best performance when we have less training samples, while the Entangled-Lasso approach performs the worst. As we increase the number of training samples, the performance of Entangled-Lasso approaches that of Disentangled-Lasso, however, learning under the Disentangled-Lasso approach is sample efficient. Disentangled-Lasso obtains $R^2$ greater than 0.5 with only 25 training samples, while other approaches obtain $R^2$ close to zero. Also, Disentagled-Lasso converges to the

optimal $R^2$ using only 50 training samples, while Entangled-Lasso does the same with 150 samples samples.

Note that the improvement due to disentanglement does not happen for the case of ridge regression as expected and there is no of a difference between the methods Disentangled-Ridge and Entangled-Ridge because the L2 norm is invariant to orthogonal transformation. Also, having sparsity in the underlying task is important. Disentangled-Lasso shows the max improvement for the case of $\ell/m = 5\%$, with the gains reducing as we decrease the sparsity in the underlying task ($l/m = 80\%$).

## D.2 SEMI-REAL EXPERIMENTS ON 3D SHAPES

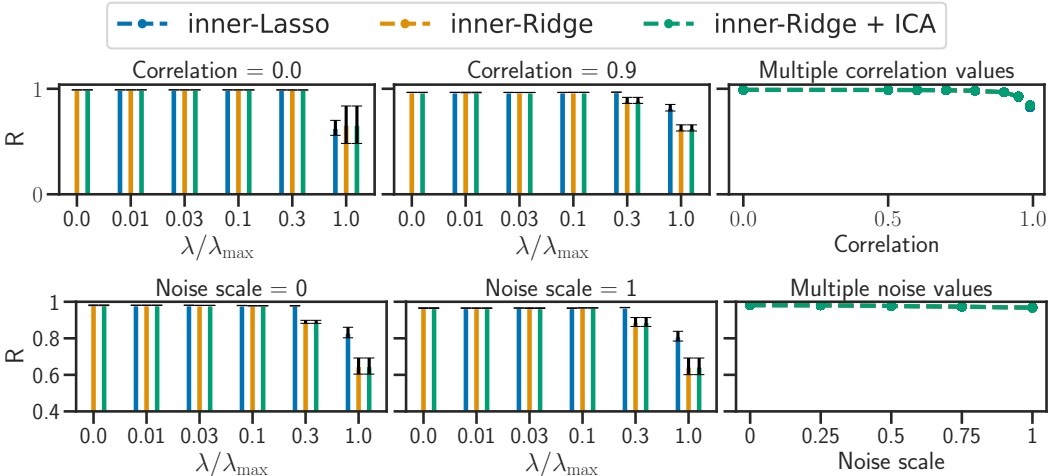

Figure 6: Prediction performance (R Score) for inner-Lasso, inner-Ridge and inner-Ridge combined with ICA as a function of the regularization parameter (left and middle). Varying level of correlation between latents (top) and noise on the latents (bottom). The right columns shows performance of the best hyperparameter for different values of correlation and noise levels.

### D.2.1 DATASET GENERATION

**Details on 3D Shapes.** The 3D Shapes dataset (Burgess & Kim, 2018) contains synthetic images of colored shapes resting in a simple 3D scene. These images vary across 6 factors: Floor hue (10 values linearly spaced in [0, 1]); Wall hue (10 values linearly spaced in [0, 1]); Object hue (10 values linearly spaced in [0, 1]); Scale (8 values linearly spaced in [0, 1]); Shape (4 values in [0, 1, 2, 3]); and Orientation (15 values linearly spaced in [-30, 30]). These are the factors we aim to disentangle. We standardize them to have mean 0 and variance 1. We denote by $\mathcal{Z} \subset \mathbb{R}^6$, the set of all possible latent factor combinations. In our framework, this corresponds to the support of the ground-truth features $\boldsymbol{f_\theta}(\boldsymbol{x})$. We note that the points in $\mathcal{Z}$ are arranged in a grid-like fashion in $\mathbb{R}^6$.

**Task generation.** For all tasks $t$, the labelled dataset $\mathcal{D}_t = \{(\boldsymbol{x}^{(t,i)}, y^{(t,i)})\}_{i=1}^n$ is generated by first sampling the ground-truth latent variables $\boldsymbol{z}^{(t,i)} := \boldsymbol{f_\theta}(\boldsymbol{x}^{(t,i)})$ i.i.d. according to some distribution $p(\boldsymbol{z})$ over $\mathcal{Z}$, while the corresponding input is obtained doing $\boldsymbol{x}^{(t,i)} := \boldsymbol{f_\theta}^{-1}(\boldsymbol{z}^{(t,i)})$ ($\boldsymbol{f_\theta}$ is invertible in 3D Shapes). Then, a sparse weight vector $\boldsymbol{w}^{(t)}$ is sampled randomly by doing $\boldsymbol{w}^{(t)} := \bar{\boldsymbol{w}}^{(t)} \odot \boldsymbol{s}^{(t)}$, were $\odot$ is the Hadamard (component-wise) product, $\bar{\boldsymbol{w}}^{(t)} \sim \mathcal{N}(\boldsymbol{0}, I)$ and $\boldsymbol{s} \in \{0,1\}^6$ is a binary vector with independent components sampled from a Bernoulli distribution with ($p = 0.5$). Then, the labels are computedfor each example as $y^{(t,i)} := \boldsymbol{w}^{(t)} \cdot \boldsymbol{x}^{(t,i)} + \epsilon^{(t,i)}$, where $\epsilon^{(t,i)}$ is independent Gaussian noise. In every tasks, the dataset has size $n = 50$. New tasks are generated continuously as we train. Figures 4 and 6 explores various choices of $p(\boldsymbol{z})$, i.e. by varying the level of correlation between the latent variables and by varying the level of noise on the ground-truth latents.

**Noise on latents.** To make the dataset slightly more realistic, we get rid of the artificial grid-like structure of the latents by adding noise to it. This procedure transforms $\mathcal{Z}$ into a new support $\mathcal{Z}_\alpha$, where $\alpha$ is the noise level. Formally, $\mathcal{Z}_\alpha := \bigcup_{z \in \mathcal{Z}}\{z + u_z\}$ where the $u_z$ are i.i.d samples from the uniform over the hypercube

$$\left[-\alpha\frac{\Delta z_1}{2}, \alpha\frac{\Delta z_1}{2}\right] \times \left[-\alpha\frac{\Delta z_2}{2}, \alpha\frac{\Delta z_2}{2}\right] \times \ldots \times \left[-\alpha\frac{\Delta z_6}{2}, \alpha\frac{\Delta z_6}{2}\right],$$

where $\Delta z_i$ denotes the gap between contiguous values of the factor $z_i$. When $\alpha = 0$, no noise is added and the support $\mathcal{Z}$ is unchanged, i.e., $\mathcal{Z}_1 = \mathcal{Z}$. As long as $\alpha \in [0, 1]$, contiguous points in $\mathcal{Z}$ cannot be interchanged in $\mathcal{Z}_\alpha$. We also clarify that the ground-truth mapping $f_\theta$ is modified to $f_{\theta, \alpha}$ consequently: for all $x \in \mathcal{X}$, $f_{\theta, \alpha}(x) := f_\theta(x) + u_z$. We emphasize that the $u_z$ are sampled only once such that $f_{\theta, \alpha}(x)$ is actually a deterministic mapping.

**Varying correlations.** To verify that our approach is robust to correlations in the latents, we construct $p(z)$ as follows: We consider a Gaussian density centered at $\mathbf{0}$ with covariance $\Sigma_{i,j} := \rho + \mathbb{1}(i = j)(1 - \rho)$. Then, we evaluate this density on the points of $\mathcal{Z}_\alpha$ and renormalize to have a well-defined probability distribution over $\mathcal{Z}_\alpha$. We denote by $p_{\alpha,\rho}(z)$ the distribution obtain by this construction.

In the top rows of Figures 4 and 6, the latents are sampled from $p_{\alpha=1,\rho}(z)$ and $\rho$ varies between 0 and 0.99. In the bottom rows of Figures 4 and 6, the latents are sampled from $p_{\alpha,\rho=0.9}(z)$ and $\alpha$ varies from 0 to 1.

### D.2.2 METRICS

We evaluate disentanglement via the *mean correlation coefficient* (Hyvärinen & Morioka, 2016; Khemakhem et al., 2020a) which is computed as follows: The Pearson correlation matrix $C$ between the ground-truth features and learned ones is computed. Then, MCC $= \max_{\pi \in \text{permutations}} \frac{1}{m} \sum_{j=1}^{m} |C_{j,\pi(j)}|$. We also evaluate linear equivalence by performing linear regression to predict the ground-truth factors from the learned ones, and report the mean of the Pearson correlations between the ground-truth latents and the learned ones. This metric is known as the *coefficient of multiple correlation*, $R$, and turns out to be the square-root of the more widely known *coefficient of determination*, $R^2$. The advantage of using $R$ over $R^2$ is that we always have MCC $\leq R$.

### D.2.3 ARCHITECTURE, INNER SOLVER & HYPERPARAMETERS

We use the four-layer convolutional neural network typically used in the disentanglement literature (Locatello et al., 2019). As mentioned in Section 2.2.2, the norm of the representation $f_{\hat{\theta}}(x)$ must be controlled to make sure the regularization remains effective. To do so, we apply batch normalization (Ioffe & Szegedy, 2015) at the very last layer of the neural network and do not learn its scale and shift parameters. Empirically, we do see the expected behavior that, without any normalization, the norm of $f_{\hat{\theta}}(x)$ explodes as we train, leading to instabilities and low sparsity.

In these experiments, the distribution $p(y; \eta)$ used for learning is a Gaussian with fixed variance. In that case, the inner problem of Section 2.2.2 reduces to Lasso regression. Computing the hypergradient w.r.t. $\theta$ requires solving this inner problem. To do so, we use Proximal Coordinate Descent (Tseng, 2001; Richtárik & Takáč, 2014).

In Figures 4 and 6, we explore various levels of regularization $\lambda$. In our implementation of inner-Lasso, $\lambda_{\max} := \frac{1}{n}\|F^\top y\|_\infty$ where $F \in \mathbb{R}^{n \times m}$ is the design matrix of the features of the samples of a task, while in the inner-Ridge implementation, $\lambda_{\max} := \frac{1}{n}\|F\|^2$.

### D.2.4 EXPERIMENTS VIOLATING ASSUMPTIONS

In this section, we explore variations of the experiments of Section 4, but this time the assumptions of Theorem 1 are violated.

Figure 7 shows different degrees of violation of Assumption 7. We consider the cases where $\mathcal{S} := \{\{1,2\},\{3,4\},\{5,6\}\}$ (block size = 2), $\mathcal{S} := \{\{1,2,3\},\{4,5,6\}\}$ (block size = 3) and $\mathcal{S} := \{\{1,2,3,4,5,6\}\}$ (block size = 6). Note that the latter case corresponds to having no sparsity

at all in the ground-truth model, i.e. all tasks requires all features. The reader can verify that these three cases indeed violate Assumption 7. In all cases, the distribution $p(S)$ puts uniform mass over its support $\mathcal{S}$. Similarly to the experiments from the main text, $\boldsymbol{w} := \bar{\boldsymbol{w}} \odot \boldsymbol{s}$, where $\bar{\boldsymbol{w}} \sim \mathcal{N}(\boldsymbol{0}, \boldsymbol{I})$ and $\boldsymbol{s} \sim p(S)$ ($\boldsymbol{s}$ is the binary representation of the set $S$). Overall, we can see that inner-Lasso does not perform as well when Assumption 7 is violated. For example, when there is no sparsity at all (block size = 6), inner-Lasso performs poorly and is even surpassed by inner-Ridge. Nevertheless, for mild violations (block size = 2), disentanglement (as measured by MCC) remains reasonably high. We further notice that all methods obtain very good R score in all settings. This is expected in light of Theorem 2, which guarantees identifiability up to linear transformation without requiring Assumption 7.

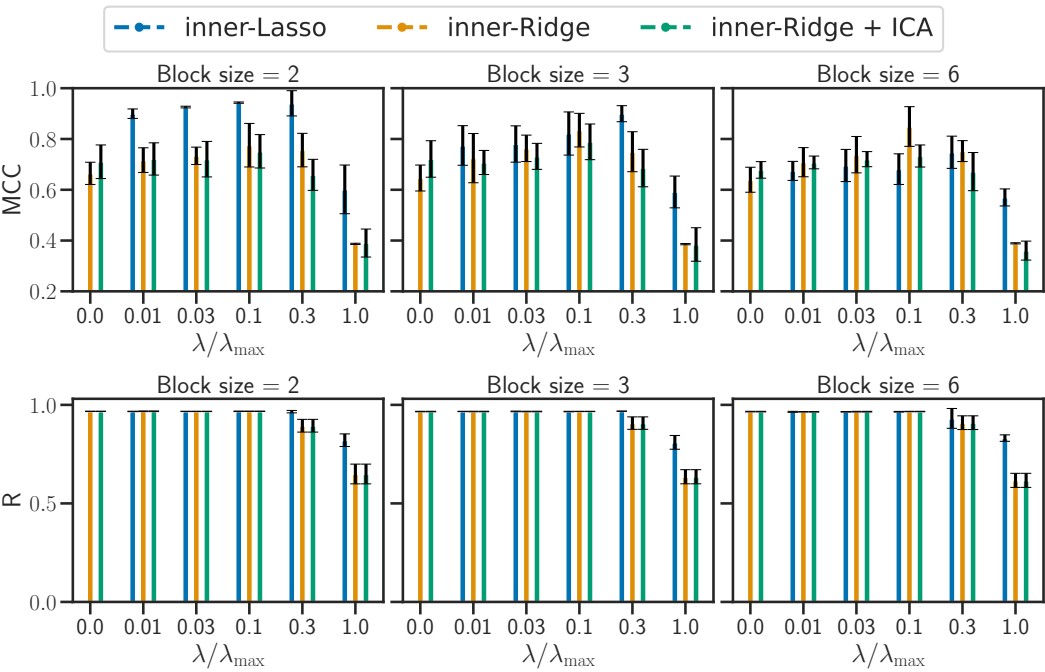

Figure 7: Disentanglement (MCC, top) and prediction (R Score, bottom) performances for inner-Lasso, inner-Ridge and inner-Ridge combined with ICA as a function of the regularization parameter. The metrics are plotted for multiple value of block size for the support. Block size = 6 corresponds to no sparsity in the ground truth coefficients.

Figure 8 presents experiments that are identical to those of Figure 4 in the main text, except for how $\boldsymbol{w}$ is generated. Here, the components of $\boldsymbol{w}$ are sampled independently according to $\boldsymbol{w}_i \sim$ Laplace($\mu = 0, b = 1$). We note that, under this process, the probability that $\boldsymbol{w}_i = 0$ is zero. This means all features are useful and Assumption 7 is violated. That being said, due to the fat tail behavior of the Laplacian distribution, many components of $\boldsymbol{w}$ will be close to zero (relatively to its variance). Thus, this can be thought of as a weaker form of sparsity where many features are relatively unimportant. Figure 8 shows that inner-Lasso can still disentangle very well. In fact, the performance is very similar to the experiments that presented actual sparsity (Figure 4).

### D.2.5 VISUAL EVALUATION

Figures 9 to 12 show how various learned representations respond to changing a single factor of variation in the image (Higgins et al., 2017, Figure 7.A.B). We see what was expected: the higher the MCC, the more disentangled the learned features appear, thus validating MCC as a good metric for disentanglement. See captions for details.

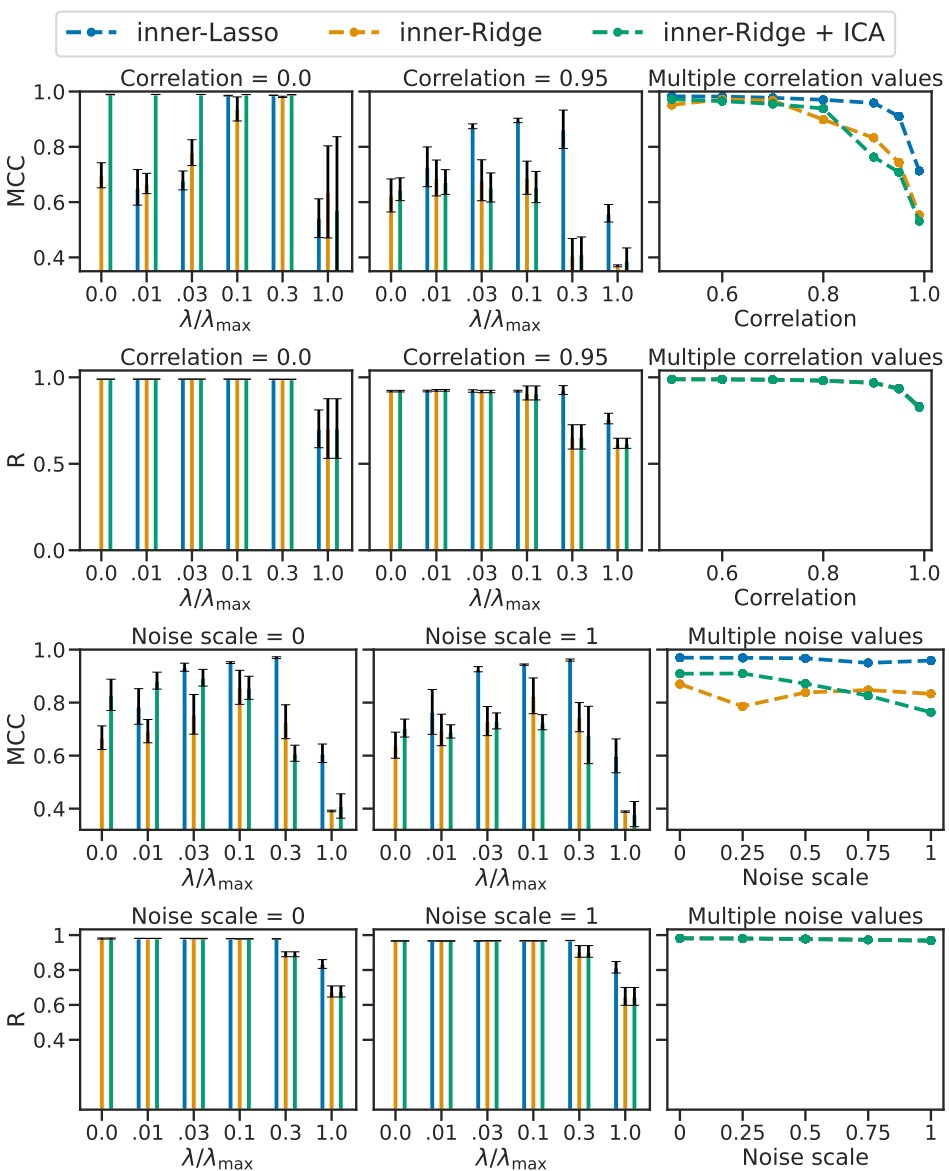

Figure 8: Same experiment as Figure 4, but the task coefficient vectors $w$ are sampled from a Laplacian distribution (instead of what was described in Appendix D.2.1). Performance is barely affected, showing some amount of robustness to violations of Assumption 7.

### D.2.6 ADDITIONAL METRICS FOR DISENTANGLEMENT

We implemented metrics from the DCI framework (Eastwood & Williams, 2018) to evaluate disentanglement. 1) DCI-Disentanglement: How many ground truth latent components are related to a particular component of the learned latent representation; 2) DCI-Completeness: How many learned latent components are related to a particular component of the ground truth latent representation. Note that for the definition of disentanglement used in the present work Definition 1, we want both DCI-disentanglement and DCI-completeness to be high.

The DCI framework requires a matrix of relative importance. In our implementation, this matrix is the coefficient matrix resulting from performing linear regression with inputs as the learned latent representation $f_{\hat{\theta}}(x)$ and targets as the ground truth latent representation $f_{\theta}(x)$, and denote the solution as the matrix $W$. Further, denote by $I = |W|$ as the importance matrix, as $I_{i,j}$ denotes the relevance of inferred latent $f_{\hat{\theta}}(x)_j$ for predicting the true latent $f_{\theta}(x)_i$.

Now, for computing DCI-disentanglement, we normalize each row of the importance matrix $I[i,:]$ by its sum so that it represents a probability distribution. Then disentanglement is given by $\frac{1}{m} \times \sum_i^m 1 - H(I[i,:])$, where $H$ denotes the entropy of a distribution. Note that for the desired case of each ground truth latent component being explained by a single inferred latent component, we would have $H(I[i,:]) = 0$ as we have a one-hot vector for the probability distribution. Similarly, for the case of each ground truth latent component being explained uniformly by all the inferred latents, $H(I[i,:])$ would be maximized and hence the DCI score would be minimized. To compute the DCI-completeness, we first normalize each column of the importance matrix $I[:,j]$ by its sum so that it represents a probability distribution and then compute $\frac{1}{m} \times \sum_i^m 1 - H(I[:,j])$.

Figure 13 shows the results for the 3D Shapes experiments (Section 4) with the DCI metric to evaluate disentanglement. Notice that we find the same trend as we had with the MCC metric 4, that inner-Lasso is more robust to correlation between the latent variables, and inner-Ridge + ICA performance drops down significantly with increasing correlation.

## D.3 META-LEARNING EXPERIMENTS

**Experimental settings.** We evaluate the performance of our meta-learning algorithm based on a group-sparse SVM base-learner on the *mini*ImageNet (Vinyals et al., 2016) dataset. Following the standard nomenclature in few-shot classification (Hospedales et al., 2021) with $k$-shot $N$-way, where $N$ is the number of classes in each classification task, and $k$ is the number of samples per class in the training dataset $\mathcal{D}_t^{\mathrm{train}}$, we consider 2 settings: 1-shot 5-way, and 5-shot 5-way. Note that the results presented in Figure 5 only show the performance on 5-shot classification. We use the same residual network architecture as in (Lee et al., 2019), with 12 layers and a representation of size $p = 1.6 \times 10^4$.

Even though we consider a similar base-learner as MetaOptNet (Lee et al., 2019) (namely, a SVM), our control experiment with $\lambda = 0$ cannot be directly compared to the performance of the model reported in that prior work. The reason is that in order to control for any other sources of "effective regularization" (e.g., data augmentation, label smoothing), we do not include the modifications made in MetaOptNet to improve performance. Moreover, we used a different solver (proximal block-coordinate descent, as opposed to a QP solver) to solve the inner problem Problem (6).

**Generalization on meta-training tasks.** In Section 2.3, we argued that evaluating the performance of the learned representations on meta-training tasks (i.e., tasks similar to the ones seen during meta-training) still shows the generalization capacity to new tasks. Indeed, those new tasks on which we evaluate performance were created using the same classes as the tasks used during meta-training, but using a combination of classes that may have not been seen in any tasks used for optimizing Problem (5). However, evaluation in meta-learning is typically done on *meta-test* tasks, i.e. tasks based on concepts that were never seen by any task during meta-training. This evaluation requires a stronger notion of generalization, closer to out-of-distribution generalization.

| Base-learner | 5-way 1-shot | 5-way 5-shot |
|---|---|---|
| SVM ($\lambda = 0$) | $53.29 \pm 0.60\%$ | $69.26 \pm 0.51\%$ |
| Group-sparse SVM ($\lambda = 0.01$) | $54.22 \pm 0.61\%$ | $70.01 \pm 0.51\%$ |
| MetaOptNet (Lee et al., 2019) | $64.09 \pm 0.62\%$ | $80.00 \pm 0.45\%$ |

Table 2: Performance of our meta-learning algorithm on the *mini*ImageNet benchmark. The performance is reported as the mean accuracy and $95\%$ confidence interval on 1000 meta-test tasks. We also report the performance of MetOptNet (Lee et al., 2019) as reference, even though the performance is not directly comparable to our SVM baseline (see text for details).

Nonetheless, we observe in Table 2 that the performance of the meta-learning method improves as the base-learners are group-sparse.

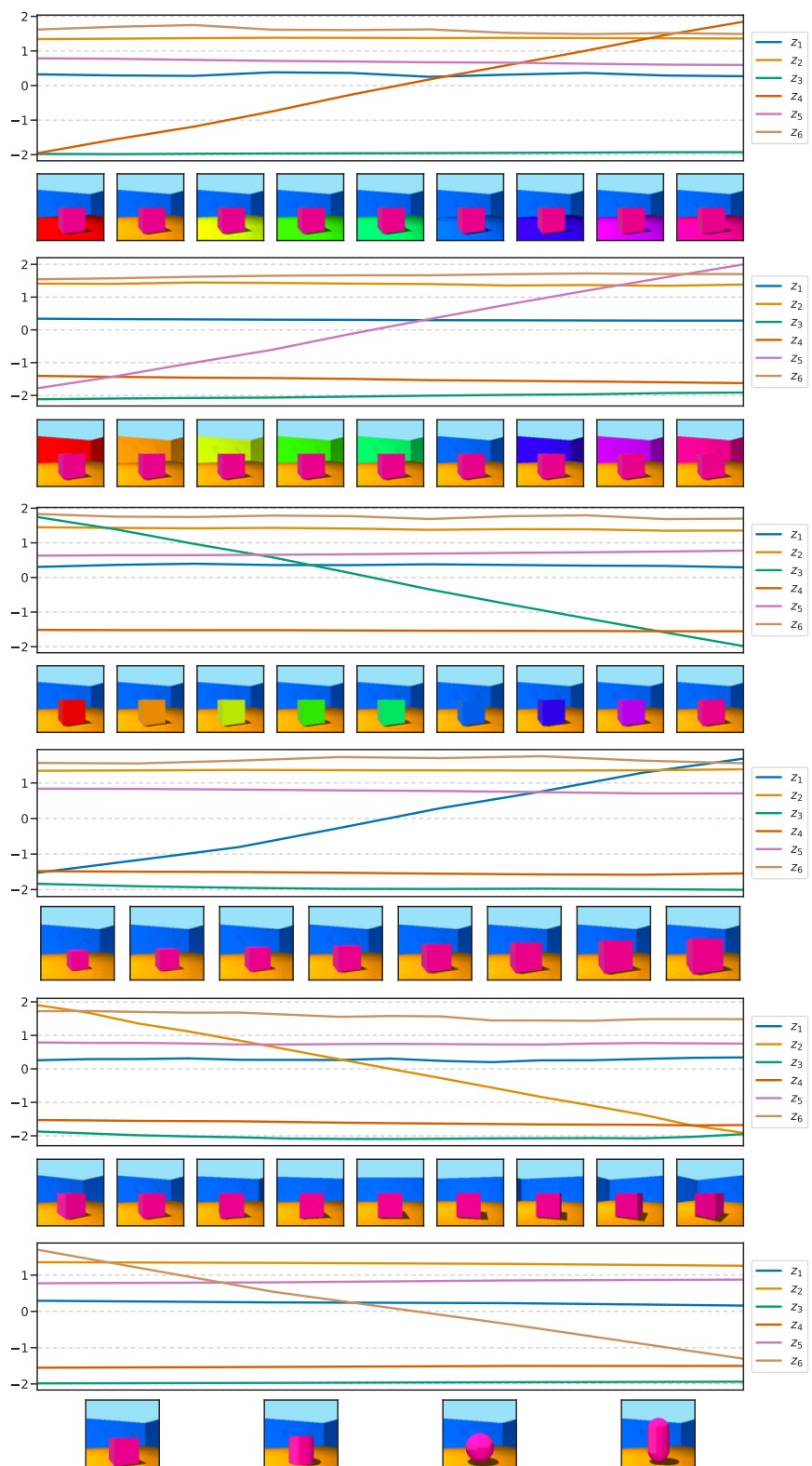

Figure 9: Varying one factor at a time in the image and showing how the learned representation varies in response. This representation was learned by **inner-Lasso** (best hyperparameter) on a dataset with **0 correlation between latents** and a noise scale of 1. The corresponding **MCC is 0.99**. We can see that varying a single factor in the image always result in changing a single factor in the learned representation.

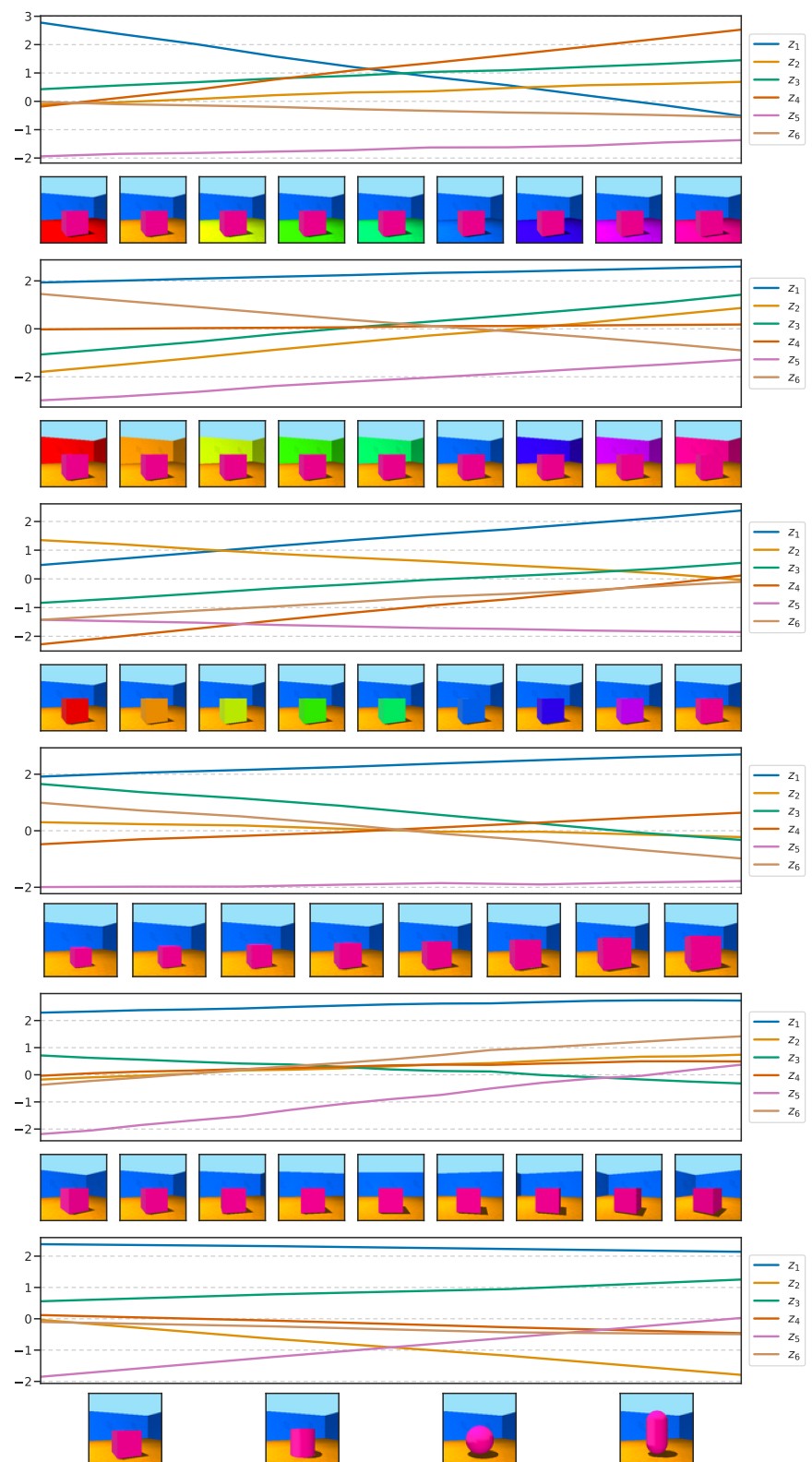

Figure 10: Varying one factor at a time in the image and showing how the learned representation varies in response. This representation was learned **without regularization** of any kind (i.e. with inner-Ridge without regularization coefficient equal to zero) on a dataset with **0 correlation between** and a noise scale of 1. The corresponding **MCC is 0.63**. We can see that varying a single factor in the image result in changing multiple factors in the learned representation, i.e. the representation is not disentangled.

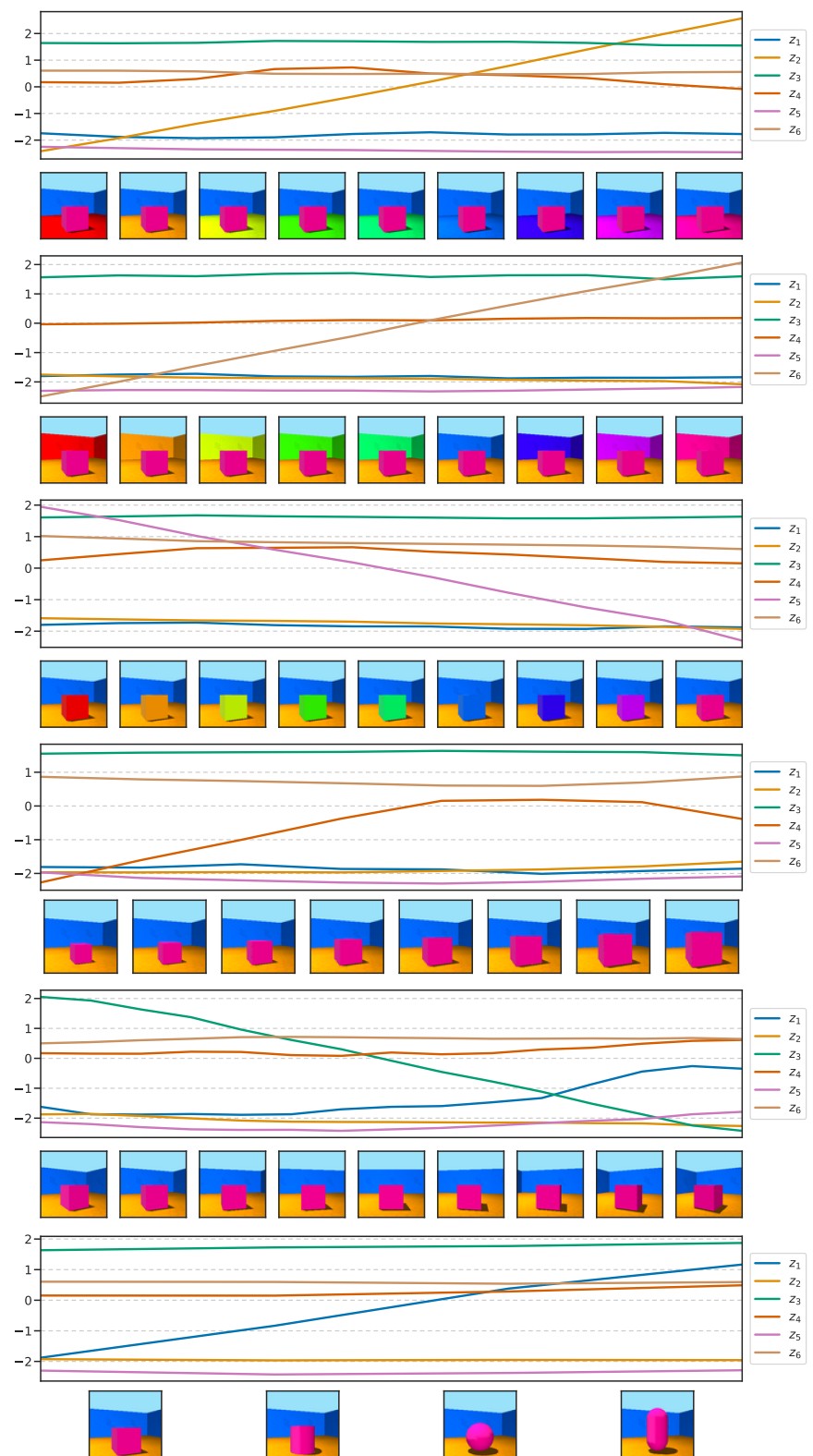

Figure 11: Varying one factor at a time in the image and showing how the learned representation varies in response. This representation was learned with **inner-Lasso** (best hyperparameter) on a dataset with **correlation 0.9 between latents** and a noise scale of 1. The corresponding **MCC is 0.96**. Qualitatively, the representation appears to be well disentangled, but not as well as in Figure 9 (reflected by a drop in MCC of 0.03).

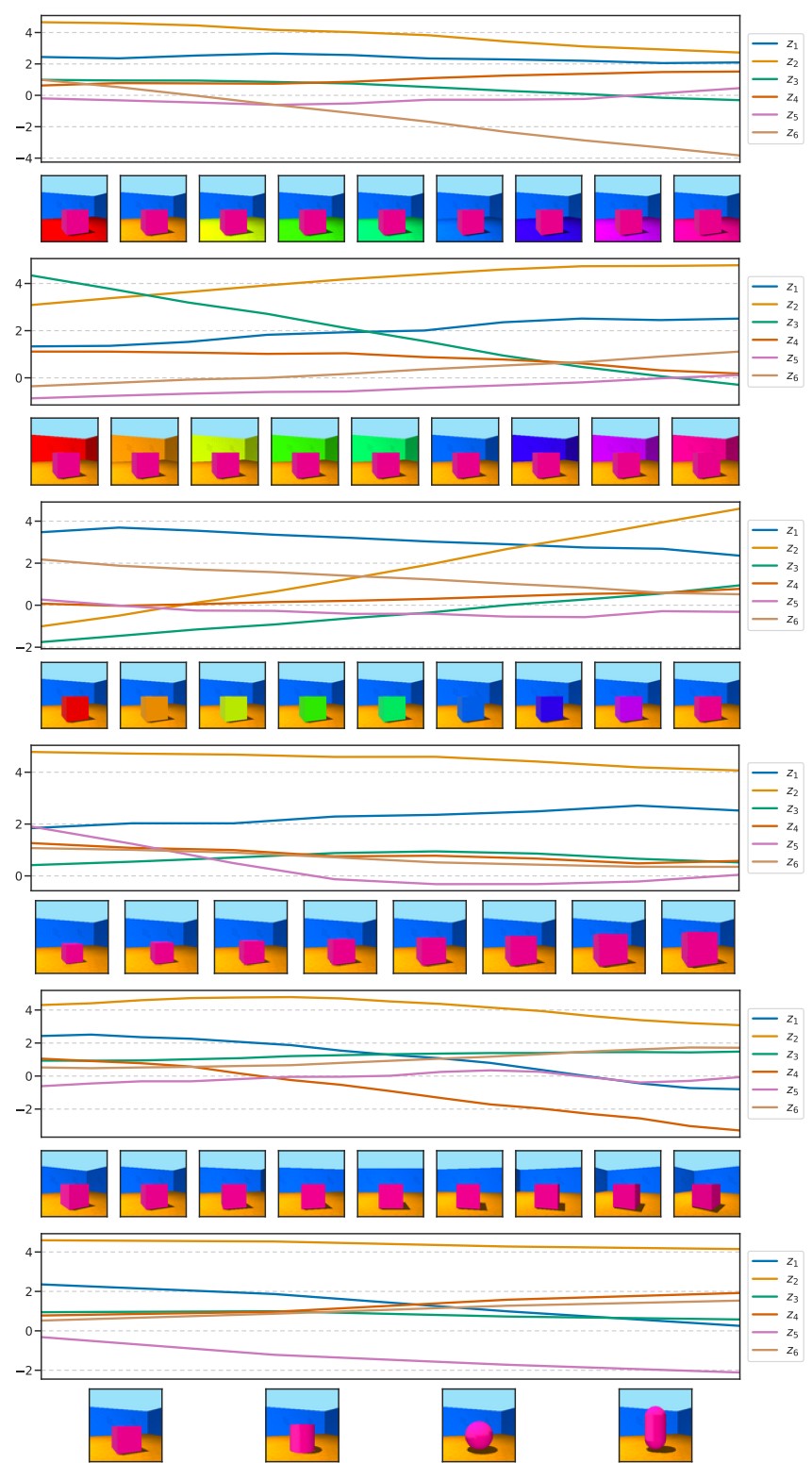

Figure 12: Varying one factor at a time in the image and showing how the learned representation varies in response. This representation was learned with **inner-Ridge** (best hyperparameter) on a dataset with **correlation 0.9 between latents** and a noise scale of 1. The corresponding **MCC is 0.79**. For most latent factors, we cannot identify a dominant feature, except maybe for background and object colors. The representation appears more disentangled than Figure 10, but less disentangled than Figure 11, as reflected by their corresponding MCC values.

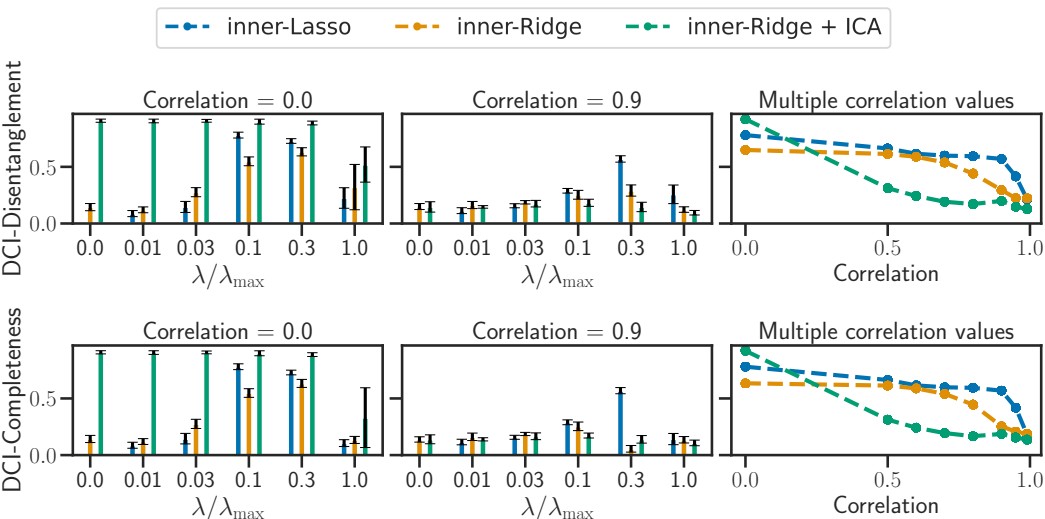

Figure 13: Disentanglement performance (DCI) for inner-Lasso, inner-Ridge and inner-Ridge combined with ICA as a function of the regularization parameter (left and middle). The right columns shows performance of the best hyperparameter for different values of correlation and noise. The top row shows the results for the disentanglement metric of DCI and the bottom row shows the results for the completeness metric of DCI.

