# OpenReview forum: "Synergies Between Disentanglement and Sparsity: a Multi-Task Learning Perspective"
_ICLR.cc/2023/Conference — Submitted to ICLR 2023_

### Official Review · Reviewer_5SbL · 2022-10-24

**Confidence:** 2
**Correctness:** 4
**Technical Novelty And Significance:** 4
**Empirical Novelty And Significance:** 4
**Recommendation:** 6

**Clarity, Quality, Novelty And Reproducibility:**

The paper is well-written, as it makes the content accessible even for someone who is not deeply familiar with the topic, as myself. Technical results and assumptions are adequately motivated and explained, and relevant experiments are presented to support the claims.
The authors provide code for reproducibility.

**Strength And Weaknesses:**

The series of technical results is definitely interesting: the authors start from the assumption that only few features of the representation are useful for prediction. Building on the assumption, they provide insight on why learning disentangled representations is beneficial to downstream performance. Finally, an algorithm is proposed to capitalize on the gained insight (and is made practical, by operating in the dual space, for the meta-learning case).
The results would appear to be adequately backed by relevant semi-synthetic experiments and on meta-learning.

It would be nice, as an add-on, if the authors could provide evidence that the proposed approach is beneficial in a larger variety of contexts.

**Summary Of The Paper:**

The authors present technical results showing that disentangled representations improve generalization if base predictors are sparse (they only use a limited number of features from the representation). Building on this, the authors propose an algorithm that learns disentangled representations (by encouraging base-predictor sparsity) and show its effectiveness on semi-synthetic data and on meta-learning.

**Summary Of The Review:**

The authors present a series of technical and empirical results that explain the practical benefit of disentangled representations if the base predictors are sparse. I am not an expert of the research area, but this appears to be an interesting and insightful paper. It would be nice to see the usefulness of the proposed approach on additional benchmarks.

---

> ### Author Response · Authors · 2022-11-18
> **Response**
>
> We thank the reviewer for the time invested in reviewing our manuscript. We were glad to read that they found that “The paper is well-written, as it makes the content accessible even for someone who is not deeply familiar with the topic…”
>
> > It would be nice, as an add-on, if the authors could provide evidence that the proposed approach is beneficial in a larger variety of contexts.
>
> The reviewer might be interested in additional experiments that we performed to assess the robustness of the proposed approach to different violations of our assumptions. For example, Figure 7 explores various degrees of violation of Assumption 7. Overall performance gradually degrades the stronger the violation is. Figure 8 reproduces experiments from the main paper, but presents tasks which are only “weakly sparse”, in the sense that all coefficients are non-zero, but many of them have values close to zero. Performance is almost unchanged, showing the method can leverage this weaker form of sparsity (which presents a violation of Assumption 7).

---

> > ### Author Response · Authors · 2022-12-08
> > **Reminder**
> >
> > This is a kind reminder that there is still time to further discuss the paper and update scores (deadline is december 12th). Did our rebuttal answered your concerns? If some questions remain, we'd be happy to provide further clarifications.

---

### Official Review · Reviewer_iXc7 · 2022-10-28

**Confidence:** 3
**Clarity, Quality, Novelty And Reproducibility:** The notations and the technical secti…
**Correctness:** 3
**Technical Novelty And Significance:** 3
**Empirical Novelty And Significance:** 3
**Recommendation:** 6

**Strength And Weaknesses:**

Strength:
The authors developed theoretical studies and results with extensive mathematical proof. The theory in the paper is relatively novel.

Weakness:
1. The experiments on disentanglement are insufficient to support the claims.  The authors need to visually show the disentanglement effects regarding different latent factors.  Otherwise, we cannot say the learned representations are meaningful.

2. Only one disentanglement metric was used in the paper, the author could use multiple disentanglement scores to validate the results.

3. The author could present additional experimental results on how disentanglement benefits prediction or other tasks, and it could make the paper stronger.

4. The notations of different variables are not easy to follow. The author could list them in a table to improve readability.


**Summary Of The Paper:**

The authors prove a new identifiability result that provides conditions under which maximally sparse base-predictors yield disentangled representations. Based on this theoretical result, they propose a practical approach to learn disentangled representations based on a sparsity-promoting bi-level optimization problem. The authors also explore a meta-learning version of the algorithm based on group Lasso multi-class SVM base predictors.

**Summary Of The Review:**

The theory developed in the paper is relatively novel. The paper could be stronger with additional experimental results on disentanglement.

---

> ### Author Response · Authors · 2022-11-18
> **Response**
>
> We thank reviewer iXc7 for providing valuable suggestions to improve our manuscript.
>
> > The experiments on disentanglement are insufficient to support the claims. The authors need to visually show the disentanglement effects regarding different latent factors.
>
> Visual inspections have been added in the experiments. As our method does not learn a generative model, we used the ‘latent response', as in Figure 7.A.B of [1]: we vary one factor at a time in the image and plot the values of the learned representation outputted by the encoder. For disentangled representations, we clearly see that only one feature changes while the others barely change. Figures 9 to 12 in Appendix D.2.5 show these latent response plots for different learned representations. As expected, the higher the MCC score, the more convincing these plots are, which confirms that MCC is a valid metric for disentanglement.
>
> >Only one disentanglement metric was used
>
> Figure 13 in the appendix now shows metrics from the DCI framework. We show DCI-disentanglement and DCI-completeness. For both metrics, the trends are similar to what we observed with MCC.
>
> > The author could list notations in a table to improve readability.
>
> Notations are now summarized in a table in the appendix.
>
> [1] Higgins, I., Matthey, L., Pal, A., Burgess, C., Glorot, X., Botvinick, M., Mohamed, S. and Lerchner, A., 2016. beta-VAE: Learning basic visual concepts with a constrained variational framework.

---

> > ### Author Response · Authors · 2022-12-08
> > **Reminder**
> >
> > This is a kind reminder that there is still time to further discuss the paper and update scores (deadline is december 12th). Did our rebuttal answered your concerns? If some questions remain, we'd be happy to provide further clarifications.

---

### Official Review · Reviewer_GZww · 2022-10-30

**Confidence:** 3
**Correctness:** 4
**Technical Novelty And Significance:** 3
**Empirical Novelty And Significance:** 3
**Recommendation:** 6

**Clarity, Quality, Novelty And Reproducibility:**

I don’t have specific concerns on the clarity. The authors wrote a well-organized draft, and reminded readers via using a lot of hyper-links.

**Strength And Weaknesses:**

Strength:

-- The paper formalizes the sparse task assumption (I.e., most tasks do not require all features), and introduces a novel identifiability result. The authors validate that disentanglement can be achieved by combining with sparse base-predictors, and can improve generalization, via both empirical evidence and the theories (under certain assumptions).

-- The theoretical analysis shed light on better understanding disentangled representations.


Overall, The authors could consider conducting experiments on more tasks and more datasets. I did realize some potential weaknesses on motivation and experiments:

-- Do the authors want to show disentanglement helps with generalization, or sparsity regularization helps with generalization? The authors can consider comparing with the disentanglement methods that do not combine with sparse base-predictors.

-- The authors propose Theorem 1 (and other theoretical results) under certain assumptions. I’m not sure how realistic those assumptions are. According to Figure one, the disentangled methods would work well under two assumptions — The ground-truth coefficient matrix (See equation 2) should be very sparse, and there should be very few training samples. Without the two conditions, combining sparsity and disentanglement do not show clear benefits.

-- It seems most theorems and propositions are based on MLE, while the meta learning part switches to SVM.
I don’t know how to understand the significance of the benefits of the group-sparse SVM method (when compared with SVM). For example, lambda/lambda_max = 0.01 yields the best performance (according to Figure 3 and Table 1 in the Appendix). In most configurations, the group-sparse solutions do not seem to outperform the vanilla SVM. In the configuration of 0.01, the benefit in terms of Meta-test is also not so big.

**Summary Of The Paper:**

The authors study how disentangled representations can be beneficial for downstream tasks under certain assumptions. The authors come up with Theorem 1, which states that leveraging multi-task learning can naturally learn disentangled representation when regularizing the base-predictors to achieve maximal-level sparsity. The authors propose to solve the problem of spare multi-task learning via solving its relaxed problem. The authors show empirical evidence on few-shot learning to support their theorem and to validate the effectiveness of their method.

**Summary Of The Review:**

Overall, I think this is an interesting paper which aims to thoroughly understand the relationship between disentanglement, sparsity and generalization. The authors tried to formalize a few intuitions, propose a few theoretical results and prove them under the dome of formalized assumptions. I think the paper shed some light on better understanding disentangled representations. It’s unclear to me how strong the theoretical results are, how good the methods would be on large realistic datasets.

---

> ### Author Response · Authors · 2022-11-18
> **Response**
>
> We would like to thank reviewer GZww for their review. We were pleased to read that they thought our “theoretical analysis shed light on better understanding disentangled representations”. We now address the main criticisms of their review.
>
> > Clarification of the goal:  disentanglement helps with generalization, or sparsity regularization helps with generalization?.
>
> Section 2.1 argues that the combination of disentanglement and sparsity regularization helps with generalization, as was mentioned in the Introduction (Section 1): “...disentangled representations coupled with sparsity-regularized base-predictors can obtain better generalization than their entangled counterparts (Section 2.1).” We also reiterate in Section 2.1: “However, disentangled representations have better generalization properties when combined with a sparse base-predictor (Proposition 2 and Figure 1).” We have added another statement at the beginning of the paragraph titled “Empirical validation” to make the point clearer.
>
> On the other hand, Section 2.2 provides disentanglement guarantees when jointly learning the representation and sparse-based predictors.
>
> > The authors can consider comparing with the disentanglement methods that do not combine with sparse base-predictors
>
> We would like to kindly clarify that the experiments show comparisons to methods that do not enforce any form of sparsity, namely inner-Ridge and inner-Ridge + ICA.
>
> > The authors propose Theorem 1 (and other theoretical results) under certain assumptions. I’m not sure how realistic those assumptions are. According to Figure 1, the disentangled methods would work well under two assumptions — The ground-truth coefficient matrix (See equation 2) should be very sparse, and there should be very few training samples. Without the two conditions, combining sparsity and disentanglement do not show clear benefits.
>
> Sparsity: We note that the assumptions of Theorem 1 allow for very sparse tasks. In fact, Assumption 7 requires some level of sparsity on the ground-truth coefficients, which, as the reviewer pointed out, is necessary for the usefulness of disentanglement, as shown in Figure 1. Thus, there is no contradiction between this assumption and the beneficial regime observed in Figure 1(this is actually the opposite).
>
> Small sample size: Theorem 1 assumes infinitely many samples, which might seem at odds with the case made in Figure 1. However, one has to distinguish between how to obtain a disentangled representation (Thm1) and how one uses a disentangled representation (Figure 1). The settings for both of these situations are typically very different. For instance, in modern representation learning frameworks, the representation is learned on a very large dataset and is used on a small labeled dataset.
>
> > It seems most theorems and propositions are based on MLE, while the meta learning part switches to SVM.
>
> The goal of the meta-learning part of the paper is to highlight a link between our bilevel formulation and state-of-the-art meta-learning algorithms. As a representative of the latter, we chose MetaOptNet, which is based on SVM classification. To have a more meaningful comparison, we came up with an SVM-based base-predictor that induces group-sparsity. In Section 2.3, we clarified that Theorem 1 is not directly applicable to our meta-learning formulation and hypothesized that the techniques developed in our work could be reused to prove a similar result for the SVM formulation.
>
> >  I don’t know how to understand the significance of the benefits of the group-sparse SVM method (when compared with SVM)
>
> Since our bi-level formulation is similar to previously studied problems in meta-learning, this experiment is to show that the proposed approach yields similar (and sometimes better) prediction performances, while using only a fraction of the features. In other words, we do not claim that our method achieves SOTA performance, but that one can learn sparse models with the same level of performance. This has been clarified in the experiment section.

---

> > ### Author Response · Authors · 2022-12-08
> > **Reminder**
> >
> > This is a kind reminder that there is still time to further discuss the paper and update scores (deadline is december 12th). Did our rebuttal answered your concerns? If some questions remain, we'd be happy to provide further clarifications.

---

### Official Review · Reviewer_wTbQ · 2022-11-04

**Confidence:** 4
**Clarity, Quality, Novelty And Reproducibility:** Please, refer to the strengths above.
**Correctness:** 4
**Technical Novelty And Significance:** 4
**Empirical Novelty And Significance:** 3
**Recommendation:** 6

**Details Of Ethics Concerns:**

No concerning ethical issues as far as can be seen.

**Strength And Weaknesses:**

++The paper is well written and motivated. The addressed problem is a problem of sufficient interest.

++The theoretical consideration of the paper is praiseworthy. Specifically, identifiability result  of Theorem 1 is particularly interesting, intuitive, and convincing.

++The paper also proposes bi-level optimization and meta-learning based approaches, which are empirically shown to be effective in Figure 1,  with fewer samples for the Disentangled-Lasso method.

++Supplementary material with additional theoretical and experimental analysis is helpful.

--The reported experiments are rather toy-like settings. It is not clear when the proposed method is expected to be useful. For example, results reported in Table.1 (supplementary material) shows that the proposed method does not compete well against MetaOptNet (although not being directly comparable).

--The meta-learning formulation of Section 3.2 is not easy to follow. The section lacks intuitive explanation and the jump form equation (7) to Proposition (3) is not well explained.

--The paper relies a lot on the supplementary material. This in itself is not a big problem, given the nature of the addressed problem. However, I believe that the paper can be made more intuitive with the help of graphic illustrations and by bringing the B.2 Discussion of Assumption in the main paper.


**Summary Of The Paper:**

This paper studies the disentangled representation learning. In particular, authors provide the evidence on the improved generalization when the disentangled representations  coupled with sparse base-predictors are learned. The paper then presents a theoretical result on identifiability condition wonder with maximally sparse base-predictors that lead to the desired disentangled representations. Relying on the presented theory, representation that maximizes sparsity is sought, in a hope of eventual disentanglement, in the context of the multi-task learning. Such sparsity maximization however not being trivial, authors propose a sparsity-promoting bi-level optimization paradigm for learning. Later, a connection between bi-lavel optimization and the meta-learning problems is established.


**Summary Of The Review:**

Overall, the paper makes a good case for the addressed problems and provides convincing and theoretical insights. The paper can benefit from more intuitive explanations (considering a broader audience) and the experiments on more realistic datasets.

---

> ### Author Response · Authors · 2022-11-18
> **Response**
>
> We would like to thank reviewer wTbQ for their review. We are pleased to know that they found that the “identifiability result of Theorem 1 is particularly interesting, intuitive, and convincing”. We now address the main points of their review.
>
> > Comparison with MetaOptNet:  For example, results reported in Table.1 (supplementary material) shows that the proposed method does not compete well against MetaOptNet (although not being directly comparable).
>
> The MetaOptNet algorithm uses (vanilla) SVMs as base-learners, but also numerous implementation tricks (including data augmentation, weight decay, drop block, label smoothing). An ablation study of these implementation tricks can be found in Table 4 of [1]. The original MetaOptNet implementation is in PyTorch, while our implementation is in Jax (which was better suited for the implicit differentiation of the non-smooth optimization problems).  We did not manage to reimplement  in Jax all the “tricks” from the pytorch implementation of MetaOptNet. We thus decided to compare the vanilla version of SVM and the group-Lasso SVM, without any tricks (within our Jax codebase).
>
> However, a recent package now allows efficient implicit differentiation in pytorch (https://github.com/metaopt/torchopt, since mid-September). We are currently finishing implementing our algorithm in pytorch using this package in order to be able to directly re-use MetaOptNet [1] pytorch code (to give an order of magnitude, MetaOptNet implementation takes 24h to run with a simple architecture). We hope to bring these experimental results before the end of the discussion deadline.
>
> > The meta-learning formulation of Section 2.3 is not easy to follow. [...] the jump from equation (7) to Proposition (3) is not well explained.
>
> Section 2.3 has been rewritten and should be clearer now. In particular, the link between the primal problem of the group Lasso soft margin SVM equation (5) (equation (7) in the previous version of the manuscript) and the dual problem (in proposition 3) is now much more detailed and better motivated. In a nutshell, proposition 3 provides the dual of problem of (5), which is simpler to solve. From the updated submission:
>
> "In few-shot learning settings, the number of features $m$ is usually much larger than the number of samples $n$ (in Lee et al. 2019, $m=1.6 10^4$ and $n\leq 25$). In such scenarios, SVMs-like problems are usually solved through their dual problems (Boyd and Vandenberghe, 2004, Chap. 5), for computational (Hsieh et al., 2008)  and theoretical (Shalev-Shwartz and Zhang 2012) benefits."
>
> > I believe that the paper can be made more intuitive with the help of graphic illustrations and by bringing the B.2 Discussion of Assumption in the main paper
>
> The discussions in Appendix B.2 have been brought in the main paper: Assumption 6 and 7 are now thoroughly discussed in the main text. In addition, these two assumptions are now visually illustrated in the main paper (Figures 2 and 3).
>
> > The paper also proposes bi-level optimization and meta-learning based approaches, which are empirically shown to be effective in Figure 1, with fewer samples for the Disentangled-Lasso method.
>
> We would like to bring a small clarification here: the experiment of Figure 1 is not obtained by solving a bi-level optimization problem. Disentangled-Lasso corresponds to performing Lasso regression on an already disentangled representation.This is now clarified in the revised manuscript: “Figure 1 compares the generalization performance of the convex relaxation of Problem (2) (Lasso regression (Tibshirani, 1996)) and Ridge regression (Hoerl & Kennard, 1970) on both disentangled and linearly entangled representations.”
>
> [1] Lee, K., Maji, S., Ravichandran, A. and Soatto, S., 2019. Meta-learning with differentiable convex optimization. In Proceedings of the IEEE/CVF conference on computer vision and pattern recognition

---

> > ### Comment · Reviewer_wTbQ · 2022-11-30
> > **Rebuttal response**
> >
> > I am happy with the authors' rebuttal. Most of my concerns are well addressed.
> > I also read other reviews and rebuttals. As I mentioned previously, this paper makes a good case and offers interesting insights.
> > Therefore, I would like to keep my original rating.

---

### Author Response · Authors · 2022-11-18
**Response to all reviewers**

We would like to thank the reviewers for the efforts they have put into reading and reviewing our manuscript. We believe their comments and suggestions have led to multiple improvements. The modifications now appear in blue in the updated pdf. We are listing here improvements which we think will be of interest to all reviewers. Note that we also answered each reviewer individually in more detail.

- Two new figures in the main text illustrating Assumption 6 and 7.
- We provided clarifications in section 2.3, which introduces the meta-learning formulation.
- Novel experiments exploring how the proposed method is robust to violations of Assumption 7.
- A visual evaluation of disentanglement in some of the learned representations in Appendix D.2.5.
- An evaluation of new metrics from the DCI framework, namely “disentanglement” and “completeness” (Appendix D.2.6), which show trends similar to the MCC metric that we used in our initial submission.
- A table summarizing notations in the appendix.

We hope the reviewers will consider updating their score if their evaluation of the manuscript changes after the discussion. If some point remains unclear, we’ll be happy to provide clarifications.

---

### Decision · Program_Chairs · 2023-01-20

**Decision:**

Reject

**Justification For Why Not Higher Score:**

Reviewers are lukewarm so hard to argue for higher.

**Justification For Why Not Lower Score:**

Good solid work.

**Metareview: Summary, Strengths And Weaknesses:**

The paper studies disentangled representations and to what degree disentanglement affect generalization in downstream tasks. Both theoretical and experimental analysis are provided.

All reviewers - despite having reservations regarding clarity of some part and the somewhat toy types setting of experiments - liked the paper and recommends acceptance. However, the reviewers have some reservations and the average score is below the typical cutoff of accepted papers so rejection is recommended. The reviewers have given plenty of constructive input that can be used to improve the paper for the next conference.